# Distinct subtypes of polycystic ovary syndrome with novel genetic associations: An unsupervised, phenotypic clustering analysis

Matthew Dapas[1], Frederick T. J. Lin[1], Girish N. Nadkarni[2], Ryan Sisk[1], Richard S. Legro[3], Margrit Urbanek[1,4,5], M. Geoffrey Hayes[1,4,6‡], Andrea Dunaif[7‡]*

**1** Division of Endocrinology, Metabolism, and Molecular Medicine, Department of Medicine, Northwestern University Feinberg School of Medicine, Chicago, Illinois, United States of America, **2** Division of Nephrology, Icahn School of Medicine at Mount Sinai, New York, New York, United States of America, **3** Department of Obstetrics and Gynecology, Penn State College of Medicine, Hershey, Pennsylvania, United States of America, **4** Center for Genetic Medicine, Northwestern University Feinberg School of Medicine, Chicago, Illinois, United States of America, **5** Center for Reproductive Science, Northwestern University Feinberg School of Medicine, Chicago, Illinois, United States of America, **6** Department of Anthropology, Northwestern University, Evanston, Illinois, United States of America, **7** Division of Endocrinology, Diabetes and Bone Disease, Icahn School of Medicine at Mount Sinai, New York, New York, United States of America

‡ These authors jointly supervised this work.
* andrea.dunaif@mssm.edu

**Data Availability Statement:** This study used data that we collected previously for our PCOS GWAS (Hayes and Urbanek et al. Nat Commun 6:7502, 2015) [19]. Stage 1 genotype data have been

## Abstract

### Background

Polycystic ovary syndrome (PCOS) is a common, complex genetic disorder affecting up to 15% of reproductive-age women worldwide, depending on the diagnostic criteria applied. These diagnostic criteria are based on expert opinion and have been the subject of considerable controversy. The phenotypic variation observed in PCOS is suggestive of an underlying genetic heterogeneity, but a recent meta-analysis of European ancestry PCOS cases found that the genetic architecture of PCOS defined by different diagnostic criteria was generally similar, suggesting that the criteria do not identify biologically distinct disease subtypes. We performed this study to test the hypothesis that there are biologically relevant subtypes of PCOS.

### Methods and findings

Using biochemical and genotype data from a previously published PCOS genome-wide association study (GWAS), we investigated whether there were reproducible phenotypic subtypes of PCOS with subtype-specific genetic associations. Unsupervised hierarchical cluster analysis was performed on quantitative anthropometric, reproductive, and metabolic traits in a genotyped cohort of 893 PCOS cases (median and interquartile range [IQR]: age = 28 [25–32], body mass index [BMI] = 35.4 [28.2–41.5]). The clusters were replicated in an independent, ungenotyped cohort of 263 PCOS cases (median and IQR: age = 28 [24–33], BMI = 35.7 [28.4–42.3]). The clustering revealed 2 distinct PCOS subtypes: a "reproductive" group (21%–23%), characterized by higher luteinizing hormone (LH) and sex hormone binding globulin (SHBG) levels with relatively low BMI and insulin levels, and a "metabolic" group

deposited in the database of Genotypes and Phenotypes (dbGaP) under the accession code phs000368.v1.p1 (https://www.ncbi.nlm.nih.gov/projects/gap/cgi-bin/study.cgi?study_id=phs000368.v1.p1). The following variables are provided along with the genotype data: SUBJID (de-identified), Case_Control (disease status), Sex, Age, Height, Weight, BMI, Race, Ethnicity. Subtype classifications for these subjects are provided in S5 Table. This study used additional array and whole-genome sequencing data from human subjects. The majority of study subjects were enrolled prior to the implementation of the NIH Genomic Data Sharing Policy in January 25, 2015. Consequently, none of the consent forms directly addressed the broad sharing of participants' data nor the risks associated with broad data sharing of these data. Further, consent forms limited the use of the DNA samples from PCOS cases to genetic analyses of this disorder. Therefore, individual-level data cannot be shared through NIH-designated repositories without approval of the Institutional Review Boards (IRBs) where the cohort was originally studied. Access to aggregate data must be limited to genetic analyses of PCOS and require approval of all relevant IRBs. Investigators may contact individual site PIs from Hayes & Urbanek et al. [19] or Kelly Brewer at kelly.brewer@mssm.edu if they are interested in collaborating on a project that requires use of quantitative trait data. The R code used to perform the clustering and subsequent family cohort classification have been uploaded to the following public GitHub repository: github.com/mdapas/PCOS_phenotype_clustering.

**Funding:** This study was supported by National Institutes of Health (NIH) Grants P50 HD044405 (AD), R01 HD057223 (AD), and R01 HD085227 (AD). MD was supported by a Ruth L. Kirschstein National Research Service Award Institutional Research Training Grant, T32 DK007169.

**Competing interests:** I have read the journal's policy and the authors of this manuscript have the following competing interests: GNN owns equity in RenalytixAI, Inc., and receives financial compensation as a consultant and advisory board member for RenalytixAI. GNN has received operational funding from Goldfinch Bio and consulting fees from BioVie Inc. and GLG consulting in the past three years. GNN is a former member of the advisory board of PulseData and received consulting fees for their services and continues to hold equity interests in PulseData.

**Abbreviations:** ACKR3, atypical chemokine receptor 3; AGMAT, agmatinase; ALK6, activin receptor-like kinase 6; AMH, anti-Müllerian hormone; ARL14EP, ADP ribosylation factor like

(37%–39%), characterized by higher BMI, glucose, and insulin levels with lower SHBG and LH levels. We performed a GWAS on the genotyped cohort, limiting the cases to either the reproductive or metabolic subtypes. We identified alleles in 4 loci that were associated with the reproductive subtype at genome-wide significance (*PRDM2/KAZN*, $P = 2.2 \times 10^{-10}$; *IQCA1*, $P = 2.8 \times 10^{-9}$; *BMPR1B/UNC5C*, $P = 9.7 \times 10^{-9}$; *CDH10*, $P = 1.2 \times 10^{-8}$) and one locus that was significantly associated with the metabolic subtype (*KCNH7/FIGN*, $P = 1.0 \times 10^{-8}$). We developed a predictive model to classify a separate, family-based cohort of 73 women with PCOS (median and IQR: age = 28 [25–33], BMI = 34.3 [27.8–42.3]) and found that the subtypes tended to cluster in families and that carriers of previously reported rare variants in *DENND1A*, a gene that regulates androgen biosynthesis, were significantly more likely to have the reproductive subtype of PCOS. Limitations of our study were that only PCOS cases of European ancestry diagnosed by National Institutes of Health (NIH) criteria were included, the sample sizes for the subtype GWAS were small, and the GWAS findings were not replicated.

## Conclusions

In conclusion, we have found reproducible reproductive and metabolic subtypes of PCOS. Furthermore, these subtypes were associated with novel, to our knowledge, susceptibility loci. Our results suggest that these subtypes are biologically relevant because they appear to have distinct genetic architecture. This study demonstrates how phenotypic subtyping can be used to gain additional insights from GWAS data.

## Author summary

### Why was this study done?

- Polycystic ovary syndrome (PCOS) is one of the most common endocrine disorders in women of reproductive age.
- The signs and symptoms of PCOS are heterogeneous, which suggests that the etiology may differ among subsets of women with PCOS.
- Elucidating the genetic mechanisms of PCOS could result in improved diagnosis and treatment.

### What did the researchers do and find?

- A clustering analysis of 893 women with PCOS, using reproductive and metabolic quantitative traits, was performed to identify subsets of affected women with similar hormonal profiles.
- There were distinct reproductive and metabolic "subtypes" of women with PCOS.
- Novel genetic variants were uniquely associated with each of the PCOS subtypes.

GTPase 14 effector protein; AS1, antisense RNA 1; BMI, body mass index; BMPR1B, bone morphogenetic protein receptor type IB; CASP9, caspase 9; CDH10, cadherin 10; CELA2, chymotrypsin like elastase 2; CI, confidence interval; COBLL1, cordon-bleu WH2 repeat protein like 1; CTRC, chymotrypsin C; C1orf195, chromosome 1 open reading frame 195; dbGaP, database of Genotypes and Phenotypes; DDI2, DNA damage inducible 1 homolog 2; DENND1A, DENN domain containing 1A; DHEAS, dehydroepiandrosterone sulfate; DNAJC16, DnaJ heat shock protein family (Hsp40) member C16; DPP4, dipeptidyl peptidase 4; EAF, effect allele frequency; EFHD2, EF-hand domain family member D2; ERG3, early growth response protein 3; ERα, estrogen receptor α; FAP, fibroblast activation protein alpha; FBLIM1, filamin binding LIM protein 1; FHAD1, forkhead associated phosphopeptide binding domain 1; FIGN, fidgetin; FLJ37453, uncharacterized LOC729614; FSH, follicle-stimulating hormone; FSHB, follicle stimulating hormone subunit beta; GCA, grancalcin; GCG, glucagon; Glu0, fasting glucose; GMM, Gaussian mixed model; GnRH, gonadotropin-releasing hormone; GRB14, growth factor receptor bound protein 14; GWAS, genome-wide association study; HA, hyperandrogenism; Hi-C, chromatin conformation capture; IFIH1, interferon induced with helicase C domain 1; Ins0, fasting insulin; IQCA1, IQ motif containing with AAA domain 1; IQR, interquartile range; KAZN, kazrin, periplakin interacting protein; KCNH7, potassium voltage-gated channel subfamily H member 7; LD, linkage disequilibrium; LH, luteinizing hormone; LOC, uncharacterized non-coding RNA; LRRC38, leucine rich repeat containing 38; MAC, minor allele count; MAF, minor allele frequency; MIR5096, microRNA 5096; MPPED2, metallophosphoesterase domain containing 2; NIH, National Institutes of Health; OD, ovulatory dysfunction; OR, odds ratio; PC, principal component; PCA, principal component analysis; PCOM, polycystic ovarian morphology; PCOS, polycystic ovary syndrome; PDPN, podoplanin; PLEKHM2, pleckstrin homology and RUN domain containing M2; PPCOSII, Pregnancy in PCOS II; PRAMEF, preferentially expressed antigen in melanoma family member; PRDM2, PR/SET domain 2; PRDM9, PR/SET domain 9; QDA, quadratic discriminant analysis; RF, random forest; RSC1A1, regulator of solute carriers 1; SHBG, sex hormone binding globulin; SLC25A34, solute carrier family 25 member 34; SLC38A11, solute carrier family 38 member 11; SNORA70F, small nucleolar RNA, H/ACA box 70F; SNP, single nucleotide polymorphism; SPEN, spen family transcriptional repressor; STREGA, Strengthening

## What do these findings mean?

- Our results suggest that there are distinct forms of PCOS that are associated with different underlying biological mechanisms.

- Women with PCOS may be poorly served by being grouped under a single diagnosis because PCOS subtypes may differ in responses to therapy and in long-term outcomes.

## Introduction

Understanding the genetic architecture of complex diseases is a central challenge in human genetics [1–3]. Often defined according to arbitrary diagnostic criteria, complex diseases can represent the phenotypic convergence of numerous genetic etiologies under the same clinical diagnosis [4–8]. Recent studies in type 2 diabetes (T2D) support the concept that there are disease subtypes with distinct genetic architecture [7,8]. Identifying and addressing genetic heterogeneity in complex diseases could increase power to detect causal variants and improve treatment efficacy [9].

Polycystic ovary syndrome (PCOS) is a highly heritable, complex genetic disorder affecting up to 15% of reproductive-age women worldwide, depending on the diagnostic criteria applied [10]. It is characterized by a variable constellation of reproductive and metabolic abnormalities [11–13]. It is the leading cause of anovulatory infertility and a major risk factor for T2D in young women [14]. Despite these substantial morbidities, the etiology (or etiologies) of PCOS remains unknown [15]. Accordingly, the commonly used diagnostic criteria for PCOS, the National Institutes of Health (NIH) criteria [16] and the Rotterdam criteria [17,18], are based on expert opinion rather than mechanistic insights and are designed to account for the diverse phenotypic presentations of PCOS. The NIH criteria require the presence of hyperandrogenism (HA) and chronic oligo/anovulation or ovulatory dysfunction (OD) [16]. The Rotterdam criteria include polycystic ovarian morphology (PCOM) and require the presence of at least 2 of these 3 key reproductive traits, resulting in 3 different affected phenotypes: HA and OD with or without PCOM, also known as NIH PCOS, as well as 2 additional non-NIH Rotterdam phenotypes, HA and PCOM and OD and PCOM.

Genome-wide association studies (GWAS) have considerably advanced our understanding of the pathophysiology of PCOS. These studies have implicated gonadotropin secretion [19] and action [20,21], androgen biosynthesis [20–22], metabolic regulation [22,23], and ovarian aging [23] in PCOS pathogenesis. A recent meta-analysis [22] of GWAS was the first study to investigate the genetic architecture of the diagnostic criteria. Only one of 14 PCOS susceptibility loci identified was significantly more strongly associated with the NIH phenotype compared to non-NIH Rotterdam phenotypes or to self-reported PCOS. These findings suggested that the genetic architecture of the phenotypes defined by the different PCOS diagnostic criteria were generally similar. Therefore, the current diagnostic criteria do not appear to identify genetically distinct disease subtypes.

It is possible to identify physiologically relevant complex disease subtypes through cluster analysis of phenotypic traits [8,24,25]. Indeed, there have been previous efforts to subtype PCOS using unsupervised cluster analysis of its hormonal and anthropometric traits [26–29]. However, there has been no validation that the resulting PCOS subtypes were biologically meaningful by testing their association with genetic variants, with other independent

the Reporting of Genetic Association Studies; SVM, support vector machine; T, testosterone; TAD, topologically associating domain; TGF-β, transforming growth factor beta; TMEM, transmembrane protein; T2D, type 2 diabetes; UNC5C, unc-5 netrin receptor C; UQCRHL, ubiquinol-cytochrome c reductase hinge protein like.

biomarkers, or with outcomes such as therapeutic responses. In this study, we sought to 1) identify phenotypic subtypes of PCOS using an unsupervised clustering approach on reproductive and metabolic quantitative traits from a large cohort of women with PCOS, 2) validate those subtypes in an independent cohort, and 3) test whether the subtypes thus identified were associated with distinct common genetic variants. As an additional validation, we investigated the association of the subtypes with rare genetic variants we recently identified in a family-based PCOS cohort [30].

## Methods

### Subjects

This study used biochemical and genotype data from our previously published PCOS GWAS, Hayes and Urbanek and colleagues [19], in which a discovery sample (Stage 1) of 984 unrelated PCOS cases and 2,964 population controls was studied, followed by a replication sample (Stage 2) of 1,799 PCOS cases and 1,231 phenotyped reproductively normal control women. All cases were of European ancestry. The present study began as an exploratory analysis to test the hypothesis that subtypes existed within the PCOS GWAS cohorts. Further analyses were performed once subtypes were identified. This study is reported according to the Strengthening the Reporting of Genetic Association Studies (STREGA) guideline (S1 Checklist). The study was approved by the Institutional Review Board of Northwestern University Feinberg School of Medicine, and each subject provided written informed consent prior to the study [19].

All PCOS cases were aged 13–45 years and were diagnosed according to the NIH criteria [10] of hyperandrogenism and chronic anovulation (8 or fewer menses per year), excluding specific disorders of the adrenal glands, ovaries, or pituitary gland [31]. Cases fulfilling the NIH criteria also meet the Rotterdam criteria for PCOS [10]. The GWAS cohorts included in the cluster analysis, the PCOS Family Study and Pregnancy in PCOS II (PPCOSII) study [19] (S1 Table), had complete data for the following traits: body mass index (BMI), testosterone (T), sex hormone binding globulin (SHBG), dehydroepiandrosterone sulfate (DHEAS), luteinizing hormone (LH), follicle-stimulating hormone (FSH), fasting insulin (Ins0), and fasting glucose (Glu0). Complete data for these quantitative traits were not available in the other GWAS cohorts because of differences in phenotyping protocols [19] (S1 Table). Two additional NIH PCOS cohorts with complete quantitative trait data were included in the present study. An ungenotyped cohort of 263 cases was used for clustering replication. A family-based whole-genome sequencing cohort of 73 PCOS cases was investigated to assess subtype clustering in families and for rare variant analysis [30].

Contraceptive steroids had been stopped at least 3 months prior to screening for the PCOS Family Study, ungenotyped, and whole-genome sequencing PCOS cohorts. Elevated T, non-SHBG bound T, and/or DHEAS levels were documented in all PCOS cases prior to enrollment in these cohorts. PPCOSII was a randomized clinical trial of letrozole versus clomiphene citrate for infertility in PCOS [32]. The PCOS cases in this study had contraceptive steroids discontinued at least 2 months prior to their baseline phenotyping visit. Since the PCOS women in this trial were seeking fertility, the majority were not on recent contraceptive steroid therapy. Therefore, it is unlikely that recent contraceptive steroid use altered T or SHBG levels in the PCOS cases included in the cluster analysis.

All subjects included in the cluster analysis were from US-based study sites. The GWAS Stage 2 replication included 2 cohorts from Europe in addition to US-based cohorts [19]. Neither European cohort was included in the cluster analysis because of incomplete quantitative trait data. We compared age and BMI in the cohorts included in the cluster analysis of cases with complete quantitative trait data versus cases from the same cohort not included because

of missing data. There were no significant differences in these parameters, suggesting that the included cases were similar to those excluded because of missing data (S2 Table).

Population-based control DNA samples for the GWAS Stage 1 sample were obtained from the NUgene biobank [33] from women of European ancestry, aged 18–97 years. Control women in the Stage 2 sample were phenotyped reproductively normal women of European ancestry, aged 15–45 years, with regular menses and normal T levels, and who were not receiving contraceptive steroids for at least 3 months prior to study [19]. T, DHEAS, SHBG, LH, FSH, Glu0, and Ins0 levels were measured as previously reported [19].

## Clustering

Clustering was performed in PCOS cases on 8 adjusted quantitative traits: BMI, T, DHEAS, Ins0, Glu0, SHBG, LH, and FSH. There were 893 combined cases from the GWAS samples with complete quantitative trait data available for clustering. Quantitative trait values were first $log_e$-normalized and adjusted for age and assay method, which varied according to the different study sites where samples were collected [19], using a linear regression. An inverse normal transformation was then applied for each trait to ensure equal scaling. The normalized trait residuals were clustered using unsupervised, agglomerative, hierarchical clustering according to a generalization of Ward's minimum variance method [34,35] on Manhattan distances between trait values. Differences in adjusted, normalized trait values between subtypes were assessed using Kruskal–Wallis and unpaired Wilcoxon rank–sum tests corrected for multiple testing (Bonferroni). Cluster stability was assessed by computing the mean Jaccard coefficient from a repeated nonparametric bootstrap resampling (n = 1,000) of the dissimilarity matrix [36]. Jaccard coefficients below 0.5 indicate that a cluster does not capture any discernable pattern within the data, while a mean coefficient above 0.6 indicates that the cluster reflects a real pattern within the data [36]. Cluster reproducibility was further assessed by repeating the clustering procedure in an independent cohort of 263 PCOS cases.

## Association testing

Stage 1 samples were genotyped using the Illumina OmniExpress (HumanOmniExpress-12v1_C; San Diego, CA, USA) single nucleotide polymorphism (SNP) array. Stage 2 samples were genotyped using the Metabochip [37] with added custom variant content based on ancillary studies and the discovery results [19]. The Stage 2 association replication in this study was therefore limited; many of the loci from Stage 1 were therefore not characterized in Stage 2. Low-quality genotypic data were removed as described previously [19]. SNPs were filtered according to minor allele frequency (MAF $\geq$ 0.01), Hardy–Weinberg equilibrium ($P \geq 1 \times 10^{-6}$), call rate ($\geq$0.99), minor allele count (MAC > 5), mendelian concordance, and duplicate sample concordance. Only autosomal SNPs were considered. Ancestry was evaluated using a principal component analysis (PCA) [38] on 76,602 linkage disequilibrium (LD)-pruned SNPs [19]. Samples with values >3 standard deviations from the median for either of the first 2 principal components (PCs) were excluded (34 in discovery; 37 in replication). Genotype data were phased using ShapeIT (v2.r790) [39] and then imputed to the 1000 Genomes reference panel (Phase3 v5) [40] using Minimac3 via the Michigan Imputation Server [41]. Imputed SNPs with an allelic $r^2$ below 0.8 were removed from analysis [42].

Association testing was performed separately for Stage 1 and Stage 2 samples. Of the 893 combined cases from both stages included in the clustering analysis, 555 were from the Stage 1 sample, and 338 were from the Stage 2 sample. In Stage 1, 2,964 normal controls were used, and 1,134 were used in Stage 2. Logistic regressions were performed using SNPTEST [43] for case–control status under an additive genetic model, adjusting for BMI and first 3 PCs of

ancestry. P-values are reported as $P_1$ and $P_2$ for Stage 1 and Stage 2, respectively. Cases were limited to specific subtypes selected from clustering results. The betas and standard errors were combined across Stage 1 and Stage 2 samples for each subtype under a fixed meta-analysis model weighting each strata by sample size [44]. Association test outputs were aligned to the same reference alleles and weighted z-scores were computed for each SNP. The square roots of each sample size were used as the proportional weights. Meta-analysis P-values ($P_{meta}$) were adjusted for genomic inflation. Associations with P-values $< 1.67 \times 10^{-8}$ were considered statistically significant, based on the standard $P < 5 \times 10^{-8}$ used in conventional GWAS adjusted for the 3 independent association tests performed.

## Chromatin interactions

Neighboring chromatin interactions were investigated in intergenic loci using high-throughput chromatin conformation capture (Hi-C) data from the 3DIV database [45]. Topologically associating domains (TADs) were identified using TopDom [46] with a window size of 20.

## Identifying subtypes in PCOS families

Quantitative trait data from the affected women (n = 73) in the family-sequencing cohort [30] were adjusted and normalized as described above. Subtype classifiers were modeled on the adjusted trait values and cluster assignments from the genotyped clustering cohort. Several classification methods were compared using 10-fold cross-validation, including support vector machine (SVM), random forest (RF), Gaussian mixed model (GMM), and quadratic discriminant analysis (QDA) [47]. The classifier with the lowest error rate was then applied to the affected women in the family-sequencing cohort to identify subtypes of PCOS in the family data. Some of the probands from the family-based cohort were included in our previous GWAS [19]. Therefore, there was some sample overlap between the training and test data: of the 893 genotyped women used to identify the original subtype clusters, 47 were also probands in the family-based cohort. Differences between subtypes in the proportion of women with *DENND1A* rare variants were tested using the chi-square test of independence.

## Results

### PCOS subtypes

Clustering was first performed in a cohort of 893 genotyped PCOS cases (Table 1, S3 Table). The clustering revealed 2 distinct phenotypic subtypes: 1) a group (23%, 207/893) characterized by higher LH and SHBG levels with relatively low BMI and Ins0 levels, which we designated "reproductive," and 2) a group (37%, 329/893) characterized by higher BMI and Glu0 and Ins0 levels with relatively low SHBG and LH levels, which we designated "metabolic" (Fig 1). The key traits distinguishing the reproductive and metabolic subtypes were BMI, insulin, SHBG, glucose, LH, and FSH, in order of importance according to relative unpaired Wilcoxon rank–sum test statistics (Fig 2). The remaining cases (40%, 357/893) demonstrated no distinguishable pattern regarding their relative phenotypic trait distributions and were designated "indeterminate" (S4 Table, S5 Table). The reproductive and metabolic subtypes clustered along opposite ends of the SHBG versus Ins0/BMI axis, which was highly correlated with the first PC of the adjusted quantitative traits (Fig 3). The reproductive subtype was the most stable cluster, with a mean bootstrapped Jaccard coefficient ($\bar{\gamma}_C$) of 0.61, followed by the metabolic subtype with $\bar{\gamma}_C = 0.55$. The indeterminate group did not appear to capture any discernable pattern within the data ($\bar{\gamma}_C = 0.41$) and was both overlapping and intermediate between the reproductive and metabolic subtypes on the SHBG versus Ins0/BMI axis.

The clustering procedure was then repeated in an independent, ungenotyped cohort of 263 NIH PCOS cases diagnosed according to the same criteria as the genotyped clustering cohort (Table 1). The clustering yielded similar results, with a comparable distribution of reproductive (26%, 68/263, $\bar{\gamma}_C = 0.57$), metabolic (39%, 104/263, $\bar{\gamma}_C = 0.46$), and indeterminate clusters (35%, 91/263, $\bar{\gamma}_C = 0.40$) (Fig 4).

## Subtype genetic associations

Genome-wide association testing was performed for each of the 3 subtypes: reproductive, metabolic, and indeterminate (Table 2). We identified alleles in 4 novel, to our knowledge, loci

**Table 1. Quantitative traits in cluster analysis PCOS cohorts by assay method.**

| Trait and Assay Method | Genotyped | | Ungenotyped | | Family Sequencing | |
|---|---|---|---|---|---|---|
| | N | Median (25–75) | N | Median (25–75) | N | Median (25–75) |
| **Age** (y) | 893 | 28 (25–32) | 263 | 28 (24–33) | 73 | 28 (25–33) |
| **BMI** (kg/m$^2$) | 893 | 35.4 (28.2–41.5) | 263 | 35.7 (28.4–42.3) | 73 | 34.3 (27.8–42.3) |
| **T** (ng/dL) | | | | | | |
| Method 1 | 620 | 72 (60–91) | 180 | 72 (61–95) | 73 | 73 (64–89) |
| Method 2 | 273 | 52 (38–69) | 83 | 65 (50–80) | – | – |
| **SHBG** (nmol/L) | | | | | | |
| Method 1 | 554 | 54 (34–81) | 176 | 55 (34–82) | 72 | 57 (38–96) |
| Method 2 | 40 | 34 (22–49) | 4 | 32 (18–54) | 1 | 37 (37–37) |
| Method 3 | 26 | 28 (18–41) | – | – | – | – |
| Method 4 | 273 | 30 (21–43) | 83 | 29 (22–48) | – | – |
| **DHEAS** (ng/mL) | | | | | | |
| Method 1 | 620 | 2,114 (1,513–2,886) | 180 | 2,190 (1,644–3,004) | 73 | 2,095 (1,509–2,774) |
| Method 2 | 273 | 1,570 (1,024–2,250) | 83 | 1,955 (1,040–2,685) | – | – |
| **Glu0** (mg/dL) | | | | | | |
| Method 1 | 192 | 90 (84–96) | 84 | 92 (88–100) | 8 | 91(87–95) |
| Method 2 | 351 | 88 (83–95) | 136 | 89 (84–95) | 48 | 91 (85–96) |
| Method 3 | 238 | 85 (77–91) | 23 | 83 (72–89) | – | – |
| Method 4 | 112 | 87 (81.5–93) | 20 | 79 (77–88) | 17 | 82 (73–88) |
| **Ins0** (µU/mL) | | | | | | |
| Method 1 | 5 | 19 (15–19) | 8 | 21 (11–57) | – | – |
| Method 2 | 614 | 22 (15–34) | 173 | 23 (16–37) | 73 | 21 (15–30) |
| Method 3 | 238 | 13 (4–21) | 23 | 13 (7–22) | – | – |
| Method 4 | 36 | 22 (15.5–30.5) | 59 | 21 (15–35) | – | – |
| **LH** (mIU/mL) | | | | | | |
| Method 1 | 515 | 12 (8–18) | 173 | 12 (7–19) | 70 | 12 (6–18) |
| Method 2 | 73 | 13 (9–17) | 7 | 11 (5–15) | 3 | 15 (12–23) |
| Method 3 | 32 | 9 (6–15) | – | – | – | – |
| Method 4 | 273 | 10 (7–14) | 83 | 10 (7–14) | – | – |
| **FSH** (mIU/mL) | | | | | | |
| Method 1 | 515 | 9 (7–11) | 173 | 10 (8–11) | 70 | 9 (8–11) |
| Method 2 | 73 | 3 (3–4) | 7 | 4 (2–5) | 3 | 3 (3–4) |
| Method 3 | 32 | 2.4 (2–3) | – | – | – | – |
| Method 4 | 273 | 6 (5–7) | 83 | 6 (5–7) | – | – |

Median trait values are shown with 25th and 75th percentiles for each clustering cohort. Details for each assay method are provided in S3 Table. **Abbreviations:** BMI, body mass index; DHEAS, dehydroepiandrosterone sulfate; FSH, follicle-stimulating hormone; Glu0, fasting glucose; Ins0, fasting insulin; LH, luteinizing hormone; N, total number; PCOS, polycystic ovary syndrome; SHBG, sex hormone binding globulin; T, testosterone.

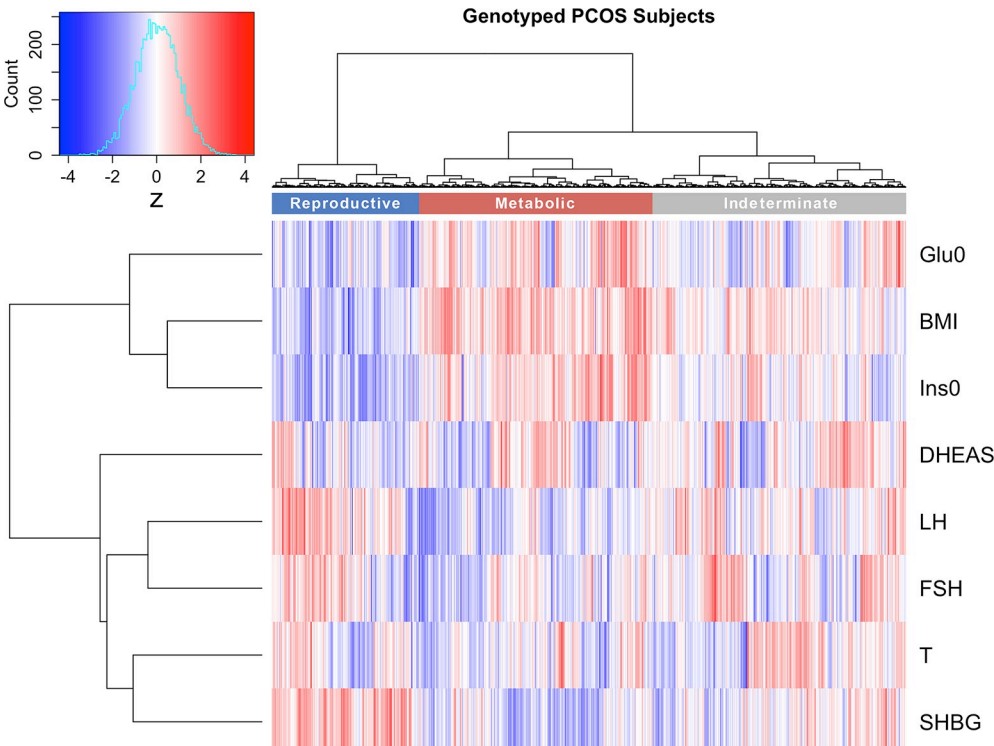

**Fig 1. Hierarchical clustering of genotyped PCOS clustering cohort.** Hierarchical clustering of 893 genotyped PCOS cases according to adjusted quantitative traits revealed 2 distinct phenotypic subtypes, a "reproductive" cluster, and a "metabolic" cluster; the remaining cases were designated as "indeterminate." The reproductive, metabolic, and indeterminate clusters are shown in the color bar as dark blue, dark red, and gray, respectively. Heatmap colors correspond to trait z-scores, as shown in the frequency histogram in which red indicates high values and blue indicates low values for each trait. The row-based dendrogram represents relative distances between trait distributions and was calculated using the same approach as the subject-based clustering. BMI, body mass index; DHEAS, dehydroepiandrosterone sulfate; FSH, follicle-stimulating hormone; Glu0, fasting glucose; Ins0, fasting insulin; LH, luteinizing hormone; PCOS, polycystic ovary syndrome; SHBG, sex hormone binding globulin; T, testosterone.

that were associated with the reproductive PCOS subtype at genome-wide significance (chromosome [chr]1 p36.21 *PRDM2/KAZN*, $P = 2.23 \times 10^{-10}$; chr2 q37.3 *IQCA1*, $P = 2.76 \times 10^{-9}$; chr4 q22.3 *BMPR1B/UNC5C*, $P = 9.71 \times 10^{-9}$; chr5 p14.2–p14.1 *CDH10*, $P = 1.17 \times 10^{-8}$) and one novel, to our knowledge, locus that was significantly associated with the metabolic subtype (chr2 q24.2–q24.3 *KCNH7/FIGN*, $P = 1.03 \times 10^{-8}$). Association testing on the indeterminate subtype replicated the 11p14.1 *FSHB* locus from our original GWAS [19] (Table 3; Figs 5 and 6).

The strongest association signal with the reproductive subtype appeared in an intergenic region of 1p36.21 579 kb downstream of the *PRMD2* gene and 194 kb upstream from the *KAZN* gene (Fig 7A). The lead SNP in the locus (rs78025940; odds ratio [OR] = 4.75, 2.82–7.98 95% confidence interval [CI], $P_1 = 2.16 \times 10^{-10}$, $P_{meta} = 2.23 \times 10^{-10}$) was imputed ($r^2 = 0.91$) in Stage 1 only. The SNP was not genotyped in Stage 2. The lead genotyped SNP in the locus (rs16850259) was also associated with the reproductive subtype with genome-wide significance ($P_{meta} = 2.14 \times 10^{-9}$) and was genotyped only in Stage 1 (OR = 5.57, 3.24–9.56 95% CI, $P_1 = 2.08 \times 10^{-9}$). In ovarian tissue, the SNPs appear to be centrally located within a large 2 Mb TAD stretching from the *FHAD1* gene to upstream of the *PDPN* gene (Fig 8).

The 2q37.3 locus spanned a 50-kb region of strong LD overlapping the 5′ end and promoter region of the *IQCA1* gene (Fig 7B). The SNP rs76182733 had the strongest association in this

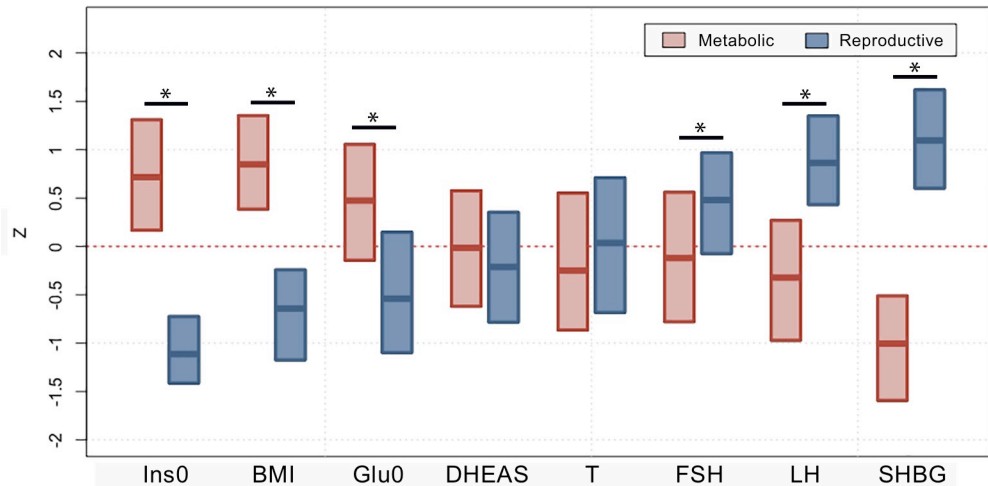

**Fig 2. Phenotypic trait distributions in reproductive and metabolic subtypes.** Median and IQRs are shown for normalized, adjusted quantitative trait distributions of genotyped PCOS cases with reproductive or metabolic subtype. The figure illustrates the traits for which the subtypes differ significantly with an asterisk (*Bonferroni adjusted Wilcoxon, $P_{adj} < 0.05$): Ins0, BMI, Glu0, FSH, LH, and SHBG. BMI, body mass index; DHEAS, dehydroepiandrosterone sulfate; FSH, follicle-stimulating hormone; Glu0, fasting glucose; Ins0, fasting insulin; IQR, interquartile range; LH, luteinizing hormone; PCOS, polycystic ovary syndrome; SHBG, sex hormone binding globulin; T, testosterone.

locus ($P_{meta} = 2.76 \times 10^{-9}$) with the reproductive subtype. The signal was genotyped only in Stage 1 (OR = 5.68, 3.00–10.78 95% CI, $P_1 = 2.69 \times 10^{-9}$) and was imputed with an imputation $r^2$ value of 0.84.

The 4q22.3 locus spanned a 500-kb region of LD, including the 3′ ends of both the *BMPR1B* and *UNC5C* genes (Fig 7C). The most strongly associated SNP (rs17023134; $P_{meta} = 9.71 \times 10^{-9}$) in the locus was within an intronic region of *UNC5C* and was associated with the reproductive subtype in the Stage 1 discovery sample with genome-wide significance (OR = 3.02, 2.06–4.42 95% CI, $P_1 = 1.40 \times 10^{-8}$) but was not significantly associated in the Stage 2 replication analysis (OR = 1.71, 0.98–2.99 95% CI, $P_2 = 7.8 \times 10^{-2}$). The SNP was imputed with an $r^2$ of 0.87 and 0.83 in the Stage 1 and Stage 2 analyses, respectively. The most strongly associated genotyped SNP in the locus (rs10516957) was also genome-wide significant ($P_{meta} = 1.46 \times 10^{-8}$) and was located in an intronic region of *BMPR1B*. The genotyped SNP was nominally associated with the reproductive subtype in both the Stage 1 (OR = 2.42, 1.66–3.52 95% CI, $P_1 = 6.72 \times 10^{-6}$) and Stage 2 (OR = 2.40, 1.51–3.82 95% CI, $P_2 = 4.7 \times 10^{-4}$) analyses with nearly identical effect sizes.

In the 5p14.2–p14.1 locus, 83 kb upstream of the *CDH10* gene (Fig 7D), 2 adjacent SNPs (rs7735176, rs16893866) in perfect LD were equally associated with the reproductive subtype with genome-wide significance ($P_{meta} = 1.17 \times 10^{-8}$). The SNPs were imputed in Stage 1 (OR = 5.09, 2.62–9.86 95% CI, $P_1 = 1.14 \times 10^{-8}$) with an imputation $r^2$ of 0.93.

The single locus containing genome-wide significant associations with the metabolic sub-type was in an intergenic region of 2q24.2–q24.3 roughly 200 kb downstream from *FIGN* and 500 kb upstream from *KCNH7* (Fig 7E). The lead SNP, rs55762028, was imputed in Stage 1 only (OR = 1.86, 0.92–3.75 95% CI, $P_1 = 9.17 \times 10^{-9}$, $P_{meta} = 1.03 \times 10^{-8}$). In pancreatic tissue, the lead SNPs appear to be located terminally within a 1.3-Mb TAD encompassing the *FIGN* gene and reaching upstream to the *GRB14* gene (Fig 9).

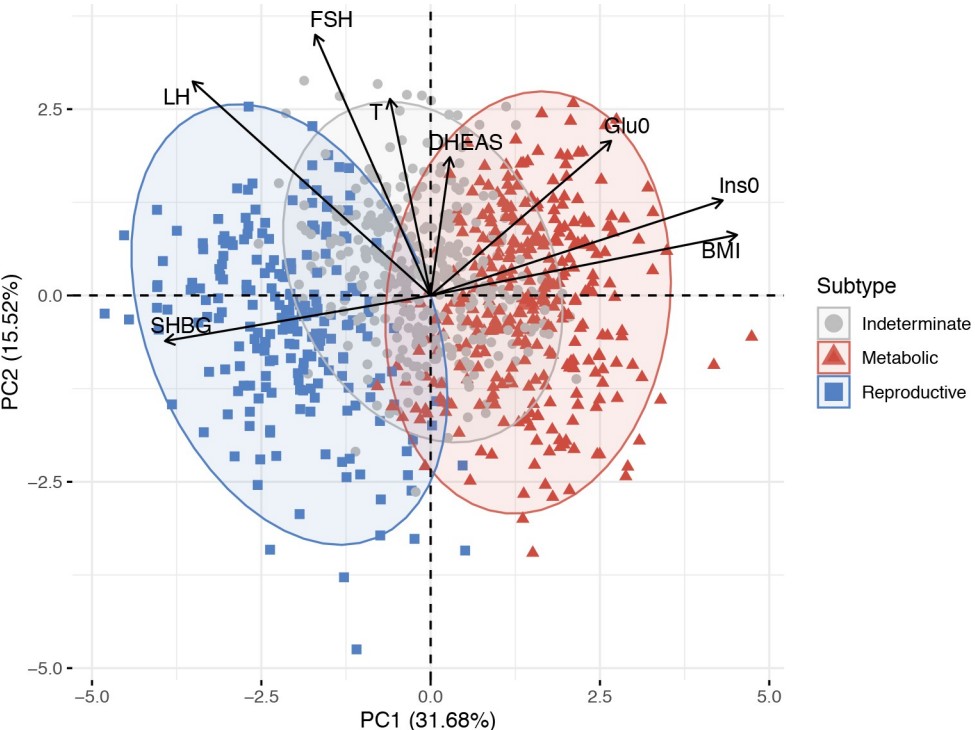

**Fig 3. PCA plot of quantitative traits for genotyped PCOS clustering cohort.** Genotyped PCOS cases are plotted on the first 2 PCs of the adjusted quantitative trait data and colored according to their identified subtype. Subtype clusters are shown as 95% concentration ellipses, assuming bivariate normal distributions. The relative magnitude and direction of trait correlations with the PCs are shown with black arrows. BMI, body mass index; DHEAS, dehydroepiandrosterone sulfate; FSH, follicle-stimulating hormone; Glu0, fasting glucose; Ins0, fasting insulin; LH, luteinizing hormone; PC, principal component; PCA, principal component analysis; PCOS, polycystic ovary syndrome; SHBG, sex hormone binding globulin; T, testosterone.

Association testing on the indeterminate cases replicated the genome-wide significant association in the 11p14.1 *FSHB* locus (Fig 7F) identified in our original GWAS (14). The lead SNP (rs10835638; $P_{meta} = 4.94 \times 10^{-12}$) and lead genotyped SNP (rs10835646; $P_{meta} = 2.75 \times 10^{-11}$) in this locus differed from the index SNPs identified in our original GWAS (rs11031006) and in the PCOS meta-analysis (rs11031005), but both of the previously identified index SNPs were also associated with the indeterminate subgroup with genome-wide significance in this study (rs11031006: $P_{meta} = 2.96 \times 10^{-10}$; rs11031005: $P_{meta} = 2.91 \times 10^{-10}$) and are in LD with the lead SNP rs10835638 ($r^2 = 0.59$) [40]. The other significant signals from our original GWAS [19] were not reproduced in any of the subtype association tests performed in this study (Table 4).

## Subtypes in PCOS families

The RF classifier yielded the lowest mean subtype misclassification rate (13.2%) compared to the SVM (13.6%), GMM (17.0%), and QDA (18.1%) models, according to 10-fold cross-validation of the genotyped clustering cohort. Affected women from the family-based cohort were classified accordingly using an RF model. Seventy-three daughters of the 83 affected women from the family-based cohort had complete quantitative trait data available for subtype classification. Seventeen (23.3%) were classified as having the reproductive subtype of PCOS, and 22 (30.1%) were classified as having the metabolic subtype. Of 14 subtyped sibling pairs, only 8

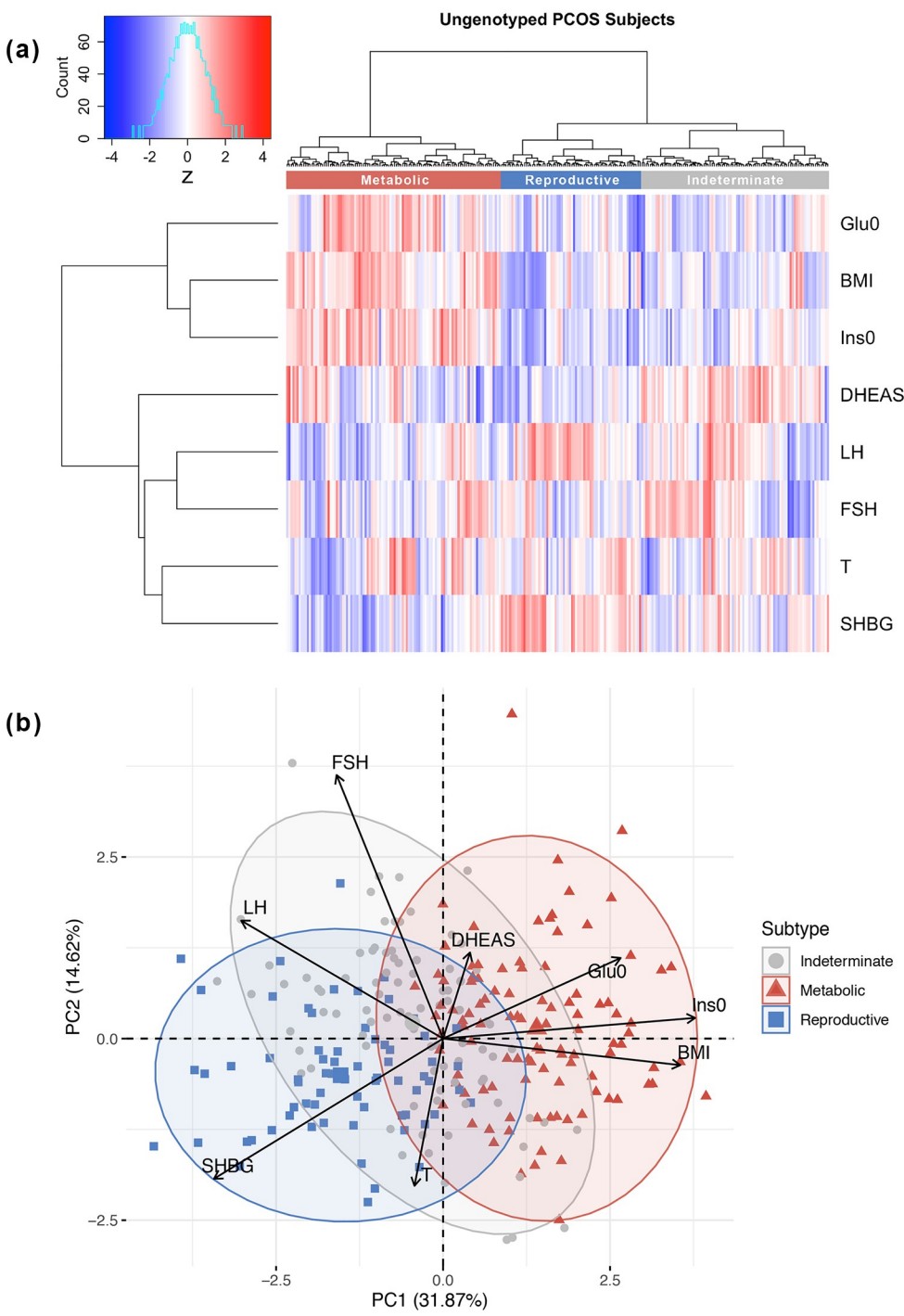

**Fig 4. Clustering of ungenotyped PCOS clustering cohort.** (a) Hierarchical clustering of 263 ungenotyped PCOS cases according to adjusted quantitative traits replicate reproductive (blue), metabolic (red), and unclassified (gray) clusters. Heatmap colors correspond to trait z-scores. (b) PCA plot of ungenotyped PCOS cases replicate results from genotyped cases. (a) Hierarchical clustering of 263 ungenotyped PCOS cases according to adjusted quantitative traits replicate reproductive (blue), metabolic (red), and indeterminate (gray) clusters. Heatmap colors correspond to trait z-scores. (b) PCA plot of ungenotyped PCOS cases replicate results from genotyped cases. BMI, body mass index; DHEAS, dehydroepiandrosterone sulfate; FSH, follicle-stimulating hormone; Glu0, fasting glucose; Ins0, fasting insulin; LH, luteinizing hormone; PC, principal component; PCA, principal component analysis; PCOS, polycystic ovary syndrome; SHBG, sex hormone binding globulin; T, testosterone.

**Table 2. Demographic characteristics of GWAS subtypes and controls.**

|  | Reproductive | Metabolic | Indeterminate | Controls |
|---|---|---|---|---|
| **N** | 207 | 329 | 357 | 4,098 |
| **Age** | 28 | 28 | 28 | 40 |
|  | (24–31) | (25–32) | (25–32) | (30–53) |
| **BMI** | 25.0 | 41.1 | 35.3 | 25.0 |
|  | (22.1–28.0) | (36.4–46.1) | (30.9–39.5) | (21.9–30.3) |

**Abbreviations:** BMI, body mass index; GWAS, genome-wide association study; IQR, interquartile range; N, total number. Data for age and BMI are expressed as median (25th–75th IQR).

were concordantly classified (57.1%); however, there was only one instance of the reproductive subtype and metabolic subtype occurring within the same nuclear family because the remaining discordant pairs each featured one indeterminate member. The proportion of affected women with one or more of the previously identified [30] deleterious, rare variants in *DENND1A* varied by subtype. Women classified as having the reproductive subtype of PCOS were significantly more likely to carry one or more of the *DENND1A* rare variants compared to other women with PCOS (*P* = 0.03; Fig 10). The distribution of affected women and *DENND1A* rare variant carriers are shown relative to the adjusted quantitative trait PCs in Fig 11.

## Discussion

It is becoming increasingly clear that common, complex traits such as T2D are a heterogeneous collection of disease subtypes [8,25,48,49]. There is emerging evidence that these subtypes have different genetic architecture [7,8,25]. Consistent with these concepts, we identified reproductive and metabolic subtypes of PCOS by unsupervised hierarchical cluster analysis of quantitative hormonal traits and BMI and found novel, to our knowledge, loci uniquely

**Table 3. Genome-wide significant associations with PCOS subtypes.**

| Chr | Mb | Variant | Gene(s) | EA | Stage 1 (Discovery) | | | | | | Stage 2 (Replication) | | | | | | $P_{meta}$ |
|---|---|---|---|---|---|---|---|---|---|---|---|---|---|---|---|---|---|
|  |  |  |  |  | EAF | β | OR | 95% CI | P | Imp $r^2$ | EAF | β | OR | 95% CI | P | Imp $r^2$ |  |
| 1 | 14.7 | rs78025940 | PRDM2/KAZN | A | 0.02 | 3.02 | 4.75 | 2.82–7.98 | $2.16 \times 10^{-10}$ | 0.91 | – | – | – | – | – | – | $2.23 \times 10^{-10}$ |
| 2 | 237.4 | rs76182733 | IQCA1 | G | 0.01 | 3.79 | 5.68 | 3.00–10.78 | $2.67 \times 10^{-9}$ | 0.84 | – | – | – | – | – | – | $2.76 \times 10^{-9}$ |
| 4 | 96.1 | rs17023134 | BMPR1B/UNC5C | G | 0.05 | 1.62 | 3.02 | 2.06–4.42 | $1.40 \times 10^{-8}$ | 0.87 | 0.06 | 0.61 | 1.71 | 0.98–2.99 | $7.81 \times 10^{-2}$ | 0.83 | $9.71 \times 10^{-9}$ |
| 5 | 24.7 | rs7735176 | CDH10 | A | 0.01 | 3.80 | 5.09 | 2.62–9.86 | $1.14 \times 10^{-8}$ | 0.93 | – | – | – | – | – | – | $1.17 \times 10^{-8}$ |
| 2 | 164.2 | rs55762028 | KCNH7/FIGN | C | 0.01 | 5.05 | 1.86 | 0.92–3.75 | $9.17 \times 10^{-9}$ | 0.96 | – | – | – | – | – | – | $1.03 \times 10^{-8}$ |
| 11 | 30.3 | rs10835638 | FSHB | T | 0.16 | 0.78 | 1.81 | 1.44–2.27 | $3.13 \times 10^{-8}$ | 0.98 | 0.17 | 0.77 | 2.01 | 1.49–2.70 | $2.67 \times 10^{-5}$ | 0.97 | $4.94 \times 10^{-12}$ |

Variant information and association statistics are shown for the most strongly associated SNP in each significant locus. Reproductive subtype loci are highlighted in blue, metabolic loci in red, indeterminate loci in gray. **Abbreviations:** Chr, chromosome; CI, confidence interval; EA, effect allele; EAF, effect allele frequency in cases and controls combined; Imp $r^2$, imputation $r^2$ for imputed SNPs; Mb, megabase position on chromosome; OR, odds ratio; P, stage-specific significance as assessed by logistic regression; PCOS, polycystic ovary syndrome; $P_{meta}$, significance as assessed by sample-size–weighted two-strata meta-analysis, adjusted for genomic inflation factor; SNP, single nucleotide polymorphism; β, effect size from association regression. Cases and controls by stage: Stage 1 = 201 metabolic, 123 reproductive, 231 indeterminate, 2,964 controls; Stage 2 = 128 metabolic, 84 reproductive, 126 indeterminate, 1,134 controls. NOTE: Not all SNPs were genotyped or imputed in both stages.

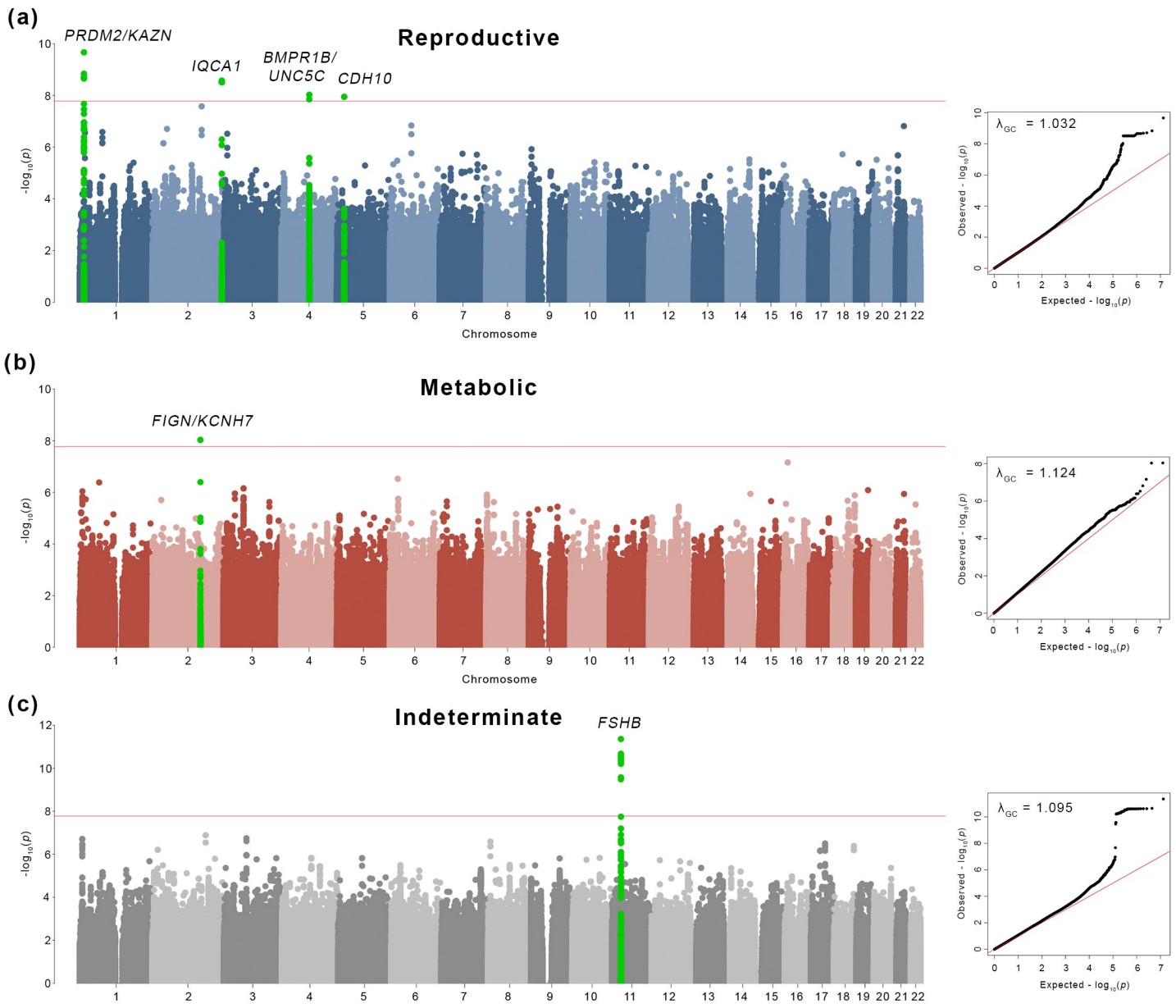

**Fig 5. Genome-wide association results.** Manhattan plots for (a) reproductive, (b) metabolic, and (c) indeterminate PCOS subtypes. The red horizontal line indicates genome-wide significance ($P \leq 1.67 \times 10^{-8}$). Genome-wide significant loci are colored in green and labeled according to nearby gene(s). Quantile–quantile plots with genomic inflation factor, $\lambda_{GC}$, are shown adjacent to corresponding Manhattan plots. PCOS, polycystic ovary syndrome.

associated with these subtypes with substantially larger effect sizes than those associated with PCOS disease status in GWAS [19–23]. We also found that a significantly greater prevalence of women classified with the reproductive subtype of PCOS carried at least one of the previously reported deleterious *DENND1A* rare variants [30] compared with those with other PCOS subtypes. These findings suggest that these subtypes are both genetically distinct as well as more etiologically homogenous [9]. Our findings are in contrast to the recent PCOS GWAS meta-analysis [22] that found that only one of 14 loci was uniquely associated with the NIH phenotype compared to non-NIH Rotterdam phenotypes. These latter findings suggest that

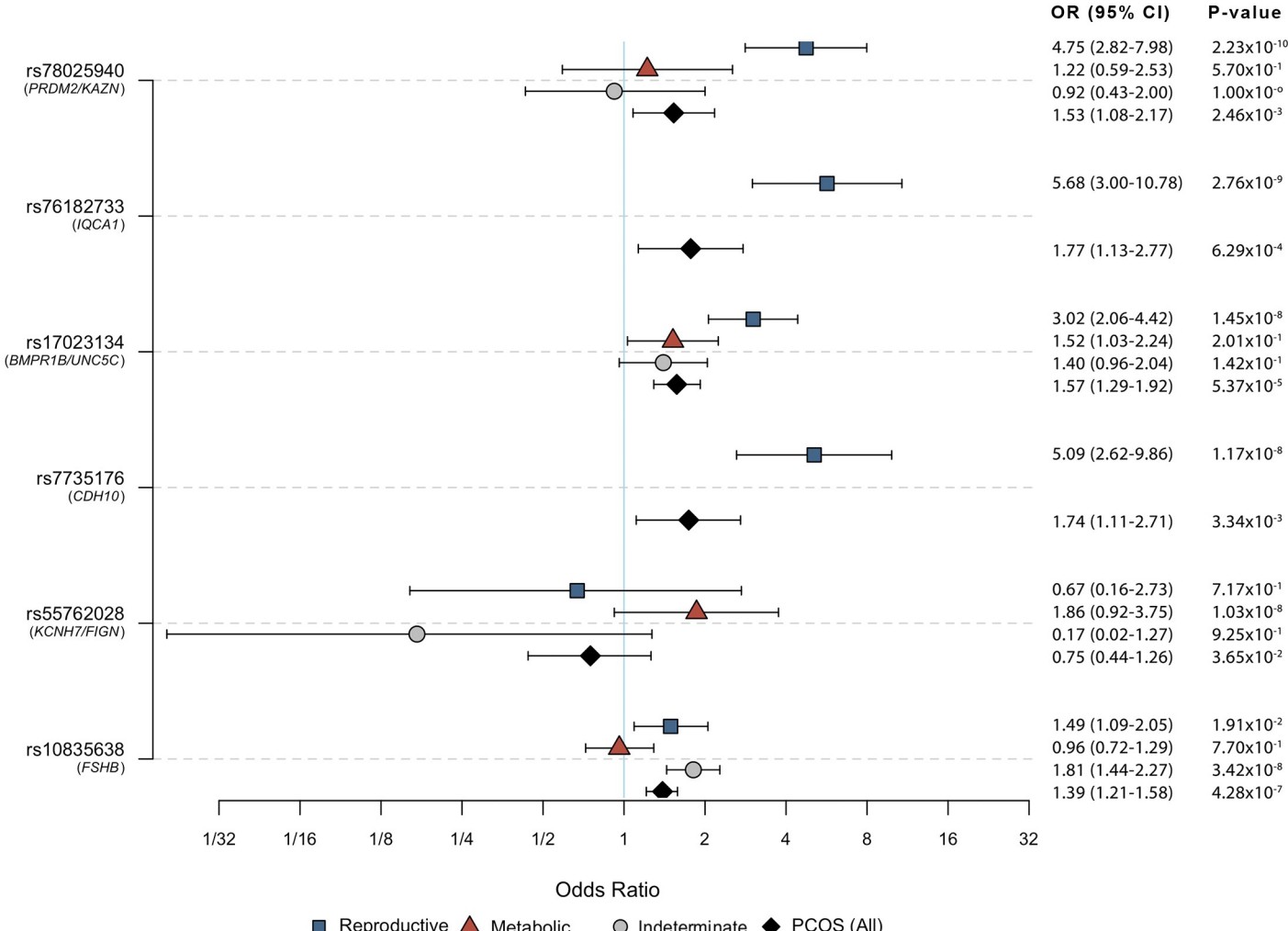

**Fig 6. Risk allele ORs in PCOS and PCOS subtypes.** ORs with 95% CIs and association P-values from the Stage 1 discovery sample are shown for each subtype-specific risk allele identified in this study relative to the corresponding values for the other subtypes and for PCOS disease status in general (includes all subtypes). Some SNPs were not characterized in certain subtypes because of low allele counts or low imputation confidence. CI, confidence interval; OR, odds ratio; PCOS, polycystic ovary syndrome; SNP, single nucleotide polymorphism.

the NIH and Rotterdam diagnostic criteria do not identify biologically distinct subtypes of PCOS. There have been previous efforts to subtype PCOS using unsupervised clustering [26–29], but no subsequent investigation into the biologic relevance of the resulting subtypes using genetic association analyses.

The key traits driving the subtypes were BMI, insulin, SHBG, glucose, LH, and FSH levels. The reproductive subtype was characterized by higher LH and SHBG levels with lower BMI and blood glucose and insulin levels. The metabolic subtype was characterized by higher BMI and glucose and insulin levels with relatively low SHBG and LH levels. The remaining 40% of cases had no distinguishable cluster-wide characteristics, and the mean trait values were between those of the reproductive and metabolic subtypes. The relative trait distributions and results of the PCAs (Figs 2, 3 and 4B) showed the reproductive and metabolic subtypes as collections of subjects on opposite ends of a phenotypic spectrum with the remaining indeterminate subjects scattered between the two. Bootstrapping and clustering in an independent cohort revealed that the reproductive and metabolic subtypes were stable and reproducible.

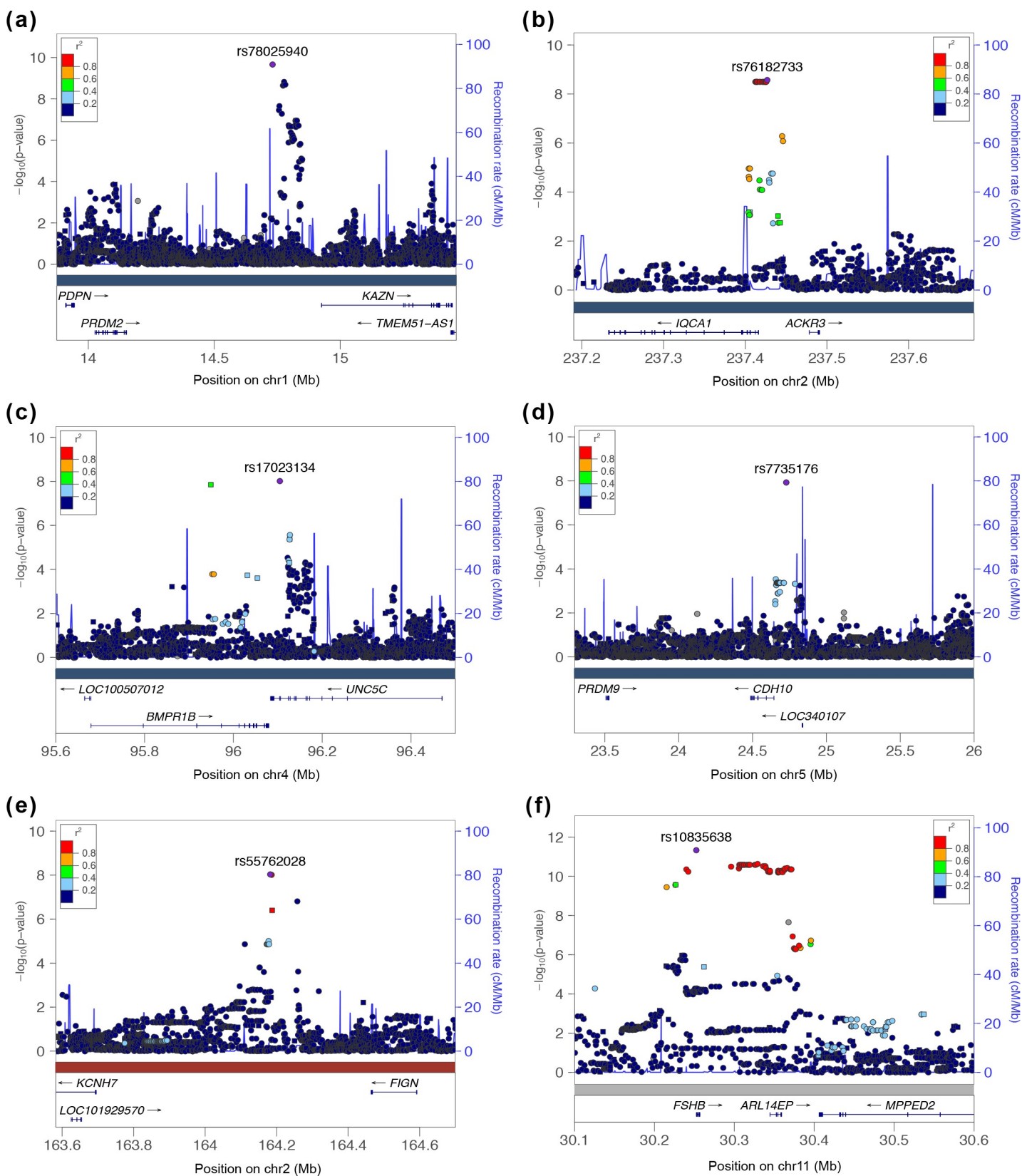

**Fig 7. Regional association plots of genome-wide significant loci.** Regional plots of association (left y-axis) and recombination rates (right y-axis) for the chromosomes (a) 1p36.21, (b) 2q37.3, (c) 4q22.3, (d) 5p14.2–p14.1, (e) 2p24.2–q24.3, and (f) 1p14.1 loci after imputation. The lead SNP in each locus is labeled and marked in purple. All other SNPs are color coded according to the strength of LD with the top SNP (as measured by r² in the European 1000 Genomes data). Imputed SNPs are plotted as circles and genotyped SNPs as squares. LD, linkage disequilibrium; SNP, single nucleotide polymorphism.

When the GWAS was repeated, different susceptibility loci were associated with the reproductive and metabolic subtypes, suggesting that they had distinct genetic architecture. The indeterminate PCOS cases were associated with the reported locus at *FSHB*, but the association

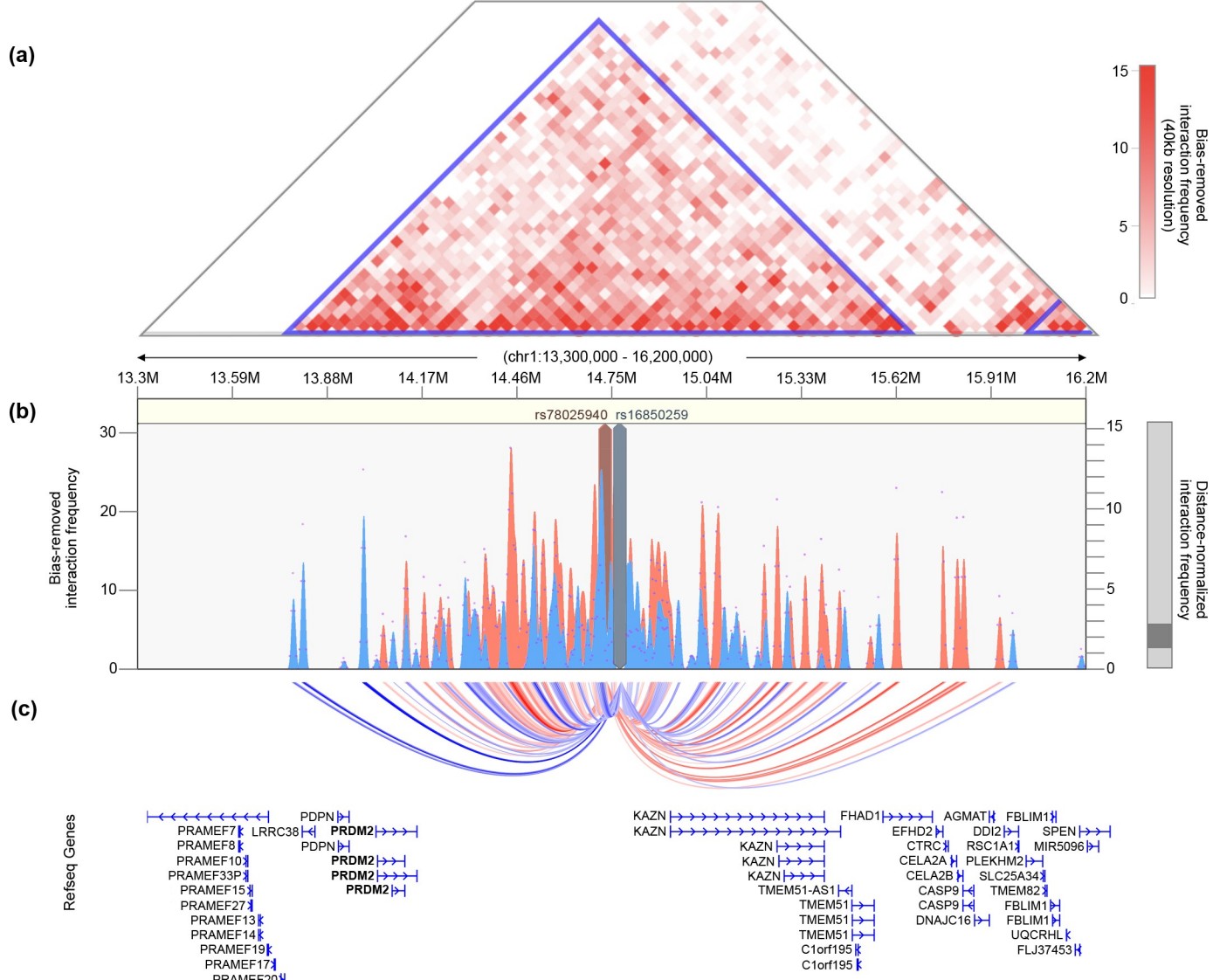

**Fig 8. Chromatin interaction map of *PRDM2/KAZN* locus.** (A) Shown is the interaction frequency heatmap from chr1:13,300,000–16,200,000 in ovarian tissue. The color of the heatmap indicates the level of normalized interaction frequencies with blue triangles indicating topological association domains. (B) One-to-all interaction plots are shown for the lead SNP (rs78025940; shown in red) and lead genotyped SNP (rs16850259; shown in blue) as bait. Y-axes on the left and the right measure bias-removed interaction frequency (red and blue bar graphs) and distance-normalized interaction frequency (magenta dots), respectively. (C) The arc representation of significant interactions for distance-normalized interaction frequencies ≥ 2 is displayed relative to the RefSeq-annotated genes in the locus. chr, chromosome; SNP, single nucleotide polymorphism.

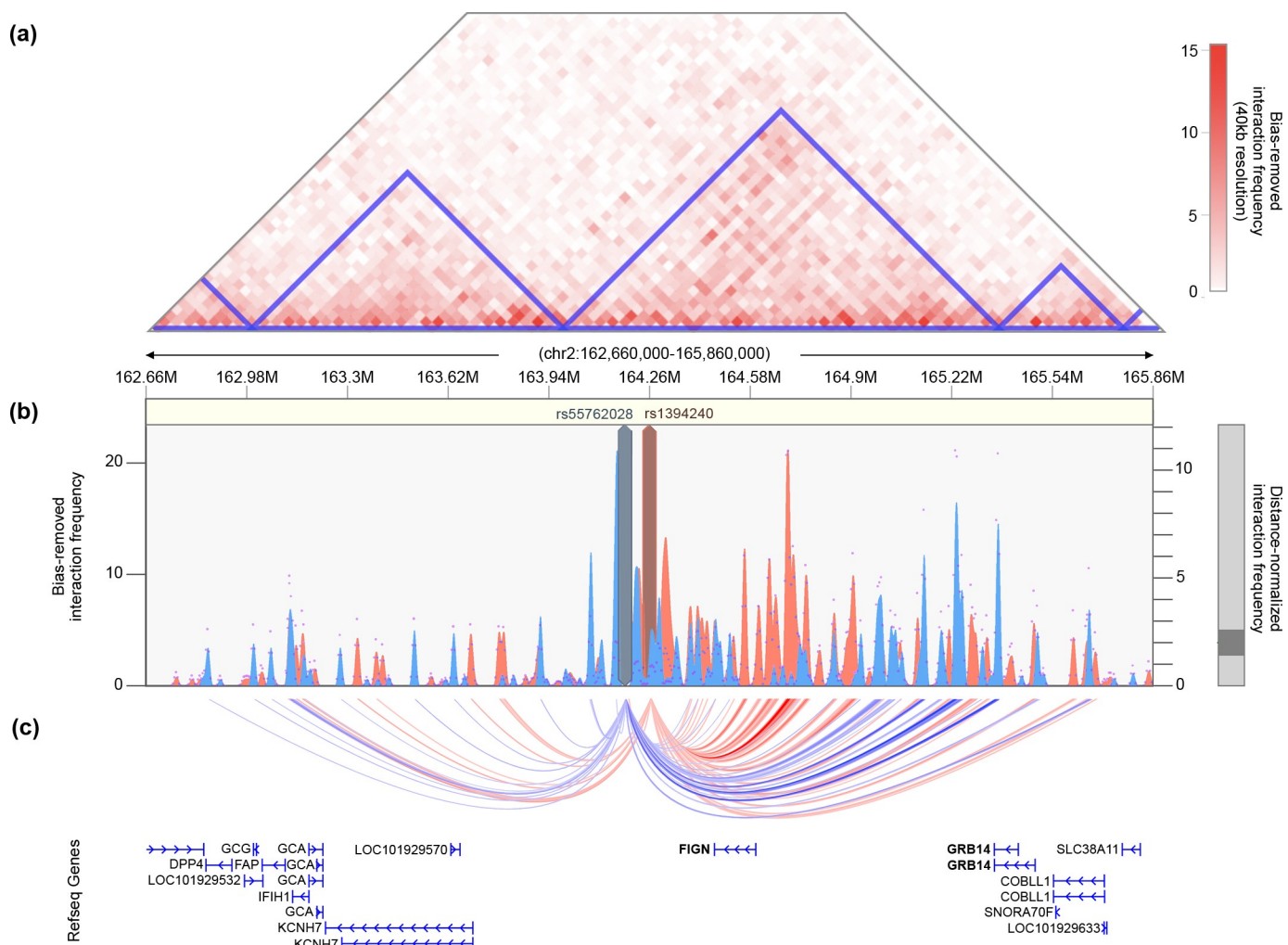

**Fig 9. Chromatin interaction map of *KCHN7/FIGN* locus.** (A) Shown is the interaction frequency heatmap from chr2:162,660,000 to 165,860,000 in pancreatic tissue. The color of the heatmap indicates the level of normalized interaction frequencies with blue triangles indicating topological association domains. (B) One-to-all interaction plots are shown for the lead SNP (rs13401392; shown in blue) and second-leading SNP (rs1394240; shown in red) as bait. Y-axes on the left and the right measure bias-removed interaction frequency (blue and red bar graphs) and distance-normalized interaction frequency (magenta dots), respectively. (C) The arc representation of significant interactions for distance-normalized interaction frequencies ≥ 2 is displayed relative to the RefSeq-annotated genes in the locus. chr, chromosome; SNP, single nucleotide polymorphism.

**Table 4. Previous GWAS association signals in PCOS subtypes.**

| Variant | Locus | PCOS | Reproductive | Metabolic | Indeterminate |
|---------|-------|------|--------------|-----------|---------------|
| rs804279 | *GATA4/NEIL2* | $P = 8.0 \times 10^{-10}$ | $P = 2.4 \times 10^{-3}$ | $P = 9.9 \times 10^{-2}$ | $P = 3.1 \times 10^{-3}$ |
| rs10993397 | *C9orf3* | $P = 4.6 \times 10^{-13}$ | $P = 2.3 \times 10^{-4}$ | $P = 6.9 \times 10^{-5}$ | $P = 1.1 \times 10^{-5}$ |
| rs11031006 | *FSHB* | $P = 1.9 \times 10^{-8}$ | $P = 8.8 \times 10^{-6}$ | $P = 6.6 \times 10^{-1}$ | $P = 3.0 \times 10^{-10}$ |

Subtype-specific association statistics are shown for each of the SNPs that were significantly associated with PCOS in Hayes and Urbanek and colleagues [19]. P = significance as assessed by sample-size–weighted two-strata meta-analysis, adjusted for genomic inflation. **Abbreviations:** GWAS, genome-wide association study; PCOS, polycystic ovary syndrome.

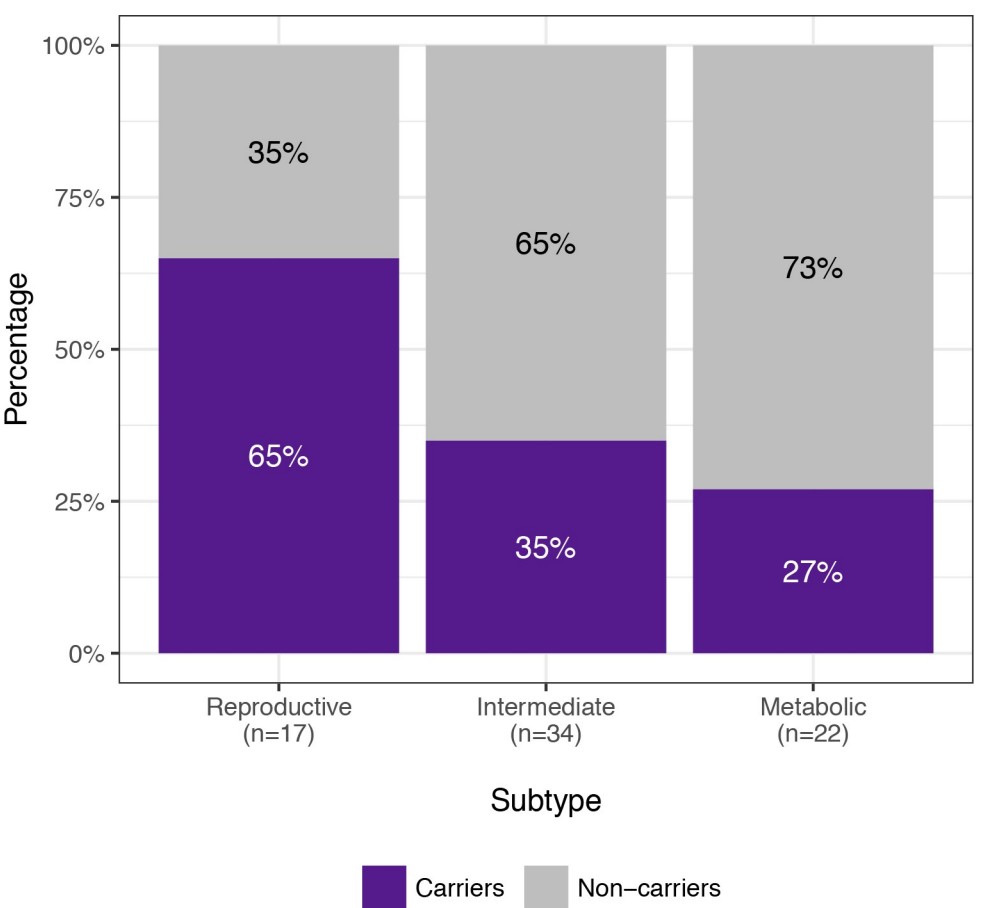

**Fig 10. *DENND1A* rare variant carriers by subtype.** The proportions of affected women with *DENND1A* rare variants in families with PCOS are shown by classified subtype. Women with the reproductive subtype were significantly more likely to carry one or more of the *DENND1A* rare variants compared to other women with PCOS (*P* = 0.03). PCOS, polycystic ovary syndrome.

signal was stronger than that of our original GWAS [19], suggesting that the indeterminate group was also more genetically homogenous after the reproductive and metabolic subtypes were removed from the analysis.

Two of the loci associated with the reproductive subtype implicate novel biologic pathways in PCOS pathogenesis. The association signal on chr1 appeared downstream of and within the same TAD as the *PRDM2* gene (Figs 7A and 8), which is an estrogen receptor coactivator [50] that is highly expressed in the ovary [51] and pituitary gland [52]. In an independent rare variant association study in PCOS families, *PRDM2* demonstrated the fifth strongest gene-level association with altered hormonal levels in PCOS families ($P = 6.92 \times 10^{-3}$) out of 339 genes tested [30]. PRDM2 binds with ligand-bound estrogen receptor alpha (ERα) to open chromatin at ERα target genes [50,53]. PRDM2 also binds with the retinoblastoma protein [54], which has been found to play an important role in follicular development in granulosa cells [55,56].

The reproductive subtype association in the 4q22.3 locus overlapped with the *BMPR1B* gene, which transcribes a type-I anti-Müllerian hormone (AMH) receptor highly expressed in granulosa cells and in gonadotropin-releasing hormone (GnRH) neurons [57] that regulates follicular development [58]. Bone morphogenetic protein receptor type IB (BMPR1B), also known as ALK6 (activin receptor-like kinase 6), heterodimerizes with the transforming growth

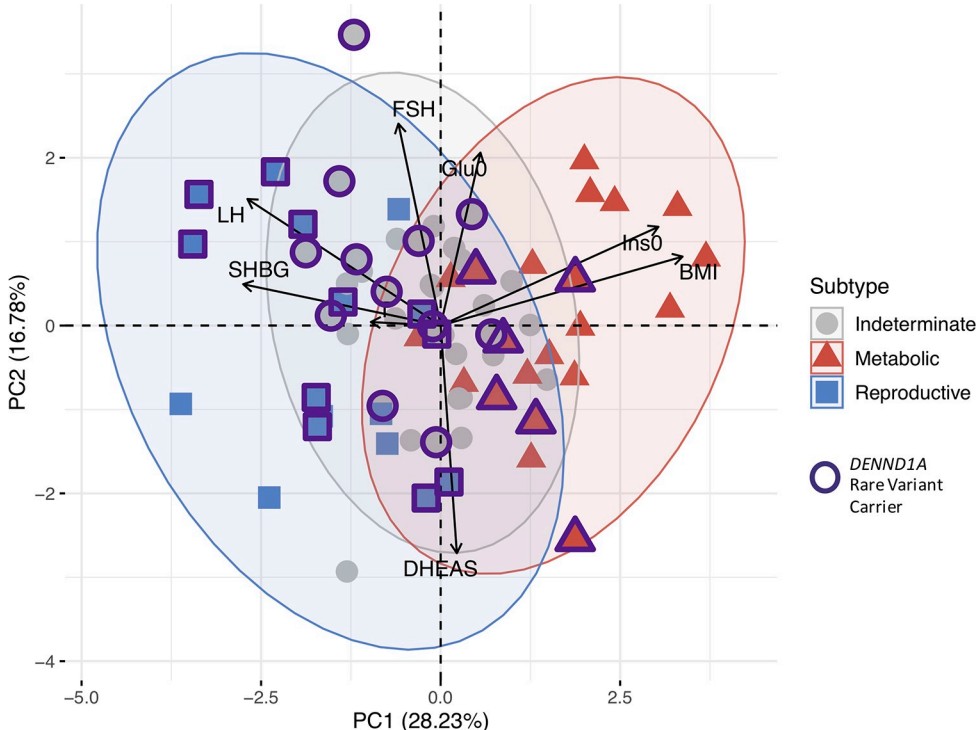

**Fig 11. PCA of affected women in PCOS families showing *DENND1A* rare variant carriers.** Affected women in PCOS families are plotted on the first 2 PCs of the adjusted quantitative trait data and colored according to their classified subtype. Markers outlined in bold represent *DENND1A* rare variant carriers. Subtype clusters are shown as 95% concentration ellipses, assuming bivariate normal distributions. The relative magnitude and direction of trait correlations with the PCs are shown with black arrows. BMI, body mass index; DHEAS, dehydroepiandrosterone sulfate; FSH, follicle-stimulating hormone; Glu0, fasting glucose; Ins0, fasting insulin; LH, luteinizing hormone; PC, principal component; PCA, principal component analysis; PCOS, polycystic ovary syndrome; SHBG, sex hormone binding globulin; T, testosterone.

factor beta (TGF-β) type-II receptors, including AMH receptor type 2 (AMHR2), and binds AMH and other BMP ligands to initialize TGF-β signaling via the Smad proteins 1, 5, and 8 [59]. *BMPR1B* has been found to mediate the AMH response in ovine granulosa cells [60], and *BMPR1B*-deficient mice are infertile and suffer from a variety of functional defects in the ovary [61,62]. One of the BMPR1B ligand genes, *BMP6*, had the third-strongest gene-level association with altered hormonal levels ($P = 4.00 \times 10^{-3}$) out of 339 genes tested in our rare variant association study in PCOS families [30]. Collectively, these results make *BMPR1B* a compelling candidate gene in PCOS pathogenesis. These findings also support our sequencing studies that have implicated pathogenic variants in the AMH signaling pathway in PCOS [63,64].

The nature of the potential involvement in PCOS is less clear for the other loci associated with the reproductive subtype. The 2q37.3 locus overlapped with the promoter region of the *IQCA1* gene. Its function in humans is not well characterized, but *IQCA1* is highly expressed in the pituitary gland [52]. The 5p14.2–p14.1 locus overlapped the promoter region of the *CDH10* gene. *CDH10* is almost exclusively expressed in the brain [51] and is putatively involved in synaptic adhesions, axon outgrowth, and guidance [65].

The lone significant association signal with the metabolic subtype was located in an intergenic region 200–280 kb downstream of the *FIGN* gene, 490–570 kb upstream of *KCNH7*. *KCNH7* encodes a voltage-gated potassium channel (subfamily H member 7, alias ERG3 [early

growth response protein 3]). *KCNH7* is primarily expressed in the nervous system [66] but has been found in murine islet cells [67,68]. *FIGN* encodes fidgetin, a microtubule-severing enzyme most highly expressed in the pituitary gland and ovary [51]. A genetic variant in *FIGN* was found to reduce the risk of congenital heart disease in Han Chinese by modulating trans-membrane folate transport [69,70]. The TAD encompassing the association signal in this locus includes *FIGN* and extends upstream to the *GRB14* gene (Fig 8). *GRB14* plays an important role in insulin receptor signaling [71,72] and has been associated with T2D in GWAS [73]. Given the various metabolic associations for the genes in this chromosomal region, it is plausi-ble that causal variants in this locus could impact a combination of these genes.

Despite evidence linking neighboring genes to PCOS pathways in each of the aforemen-tioned loci, it remains possible, of course, that other, more distant genes in LD underlie the association signals. Causal variants are often up to 2 Mb away from the associated SNP, not necessarily in the closest gene [74]. Fine-mapping and functional studies are needed in order to confirm the causal variants in each of these loci. In addition, the sample sizes for the subtype GWAS were small, some of the associations were based only on imputed SNPs in Stage 1, and a replication association study has not yet been performed. However, the aforementioned functional evidence for several of the loci—particularly for *PRDM2* and *BMPR1B*—support the validity of their associations. Further, the fact that one of the genes associated with the reproductive subtype, *PRDM2*, was associated with PCOS quantitative traits in our family-based analysis [30] does represent a replication of this signal by an independent analytical approach. Nevertheless, our genetic association results should be considered preliminary.

The effect sizes of the subtype alleles, particularly those associated with the reproductive subtype (OR 3.02–5.68) (Table 3), were substantially greater than the effects (OR 0.70–1.51) observed for alleles associated with PCOS diagnosis in previous GWAS [19–23]. In general, there is an inverse relationship between allele frequency and effect size [1] because alleles with larger phenotypic effects are subject to purifying selection and, therefore, occur less frequently in the population [75,76]. Accordingly, in contrast to the common variants (effect allele fre-quency [EAF] > 0.05) associated with PCOS in previous GWAS [19–23], the alleles associated with the subtypes were all of low frequency (EAF 0.01–0.05; Table 3). However, given the lim-ited cohort sizes in this study, the subtype association testing did not have adequate power to detect associations with more modest effect sizes, such as those from our previous GWAS [19]. It is also possible that the large effect sizes were somewhat inflated by the so-called "winner's curse" [77,78], but they nonetheless suggest that the subtypes were more genetically homoge-neous than PCOS diagnosis in general.

In applying a subtype classifier to our family-based cohort, we found 12 affected sibling pairs in which at least one of the daughters was classified with the reproductive or metabolic subtype. Six of these sibling pairs were classified with the same subtype. There was only one discordant pairing of the reproductive subtype with the metabolic subtype. This further sug-gests that the reproductive and metabolic subtypes are genetically distinct in their origins. The greater prevalence of *DENND1A* rare variant carriers observed in women with the reproduc-tive subtype in the family-based cohort implicates this gene in the pathogenesis of this subtype. *DENND1A* is known to regulate androgen biosynthesis in the ovary [79,80]; therefore, we would expect *DENND1A*-mediated PCOS to be more closely associated with the reproductive subtype of PCOS. However, we did not find an association between any *DENND1A* alleles and the reproductive subtype in the subtype GWAS, perhaps because of allelic heterogeneity or our limited power to detect associations with more modest effect sizes.

We only studied women with PCOS as defined by the NIH diagnostic criteria. Future stud-ies will investigate whether similar reproductive and metabolic clusters are present in non-NIH Rotterdam PCOS cases. In particular, it is possible that there will be no metabolic subtype

in non-NIH Rotterdam PCOS cases because these phenotypes have minimal metabolic risk [81,82]. Indeed, in a previous effort to identify phenotypic subtypes in Rotterdam PCOS cases [29], the cluster that most closely resembled the reproductive subtype represented the largest proportion of PCOS women at 44%, of whom only 78% met the NIH criteria for PCOS, whereas the cluster that most closely resembled the metabolic subtype constituted only 12% of the total sample, but 98% met the NIH diagnostic criteria. Furthermore, trait distributions may vary among women with PCOS from different geographic locations, such as in some of the sites excluded from our analysis because of incomplete quantitative trait data. For example, European PCOS cases have a lower prevalence of obesity compared to US PCOS cases [83]. Because of the within-cohort normalization of quantitative traits prior to clustering, our method is well-suited for identifying subsets of cases within populations, but therefore, it may not be suitable for directly comparing subtype membership between populations.

Our clustering cohorts included only US-based women of European ancestry. It will be of considerable importance to investigate whether subtypes are present in women with PCOS of other ancestries and geographic regions. Women with PCOS of diverse races and ethnicities have similar reproductive and metabolic features [84–86]. However, there are differences in the severity of the metabolic defects due to differences in the prevalence of obesity [83], as well as to racial/ethnic differences in insulin sensitivity [87,88]. Furthermore, the susceptibility loci associated with subtypes in other ancestry groups may differ because the low frequency and large effect size of the variants associated with the reproductive subtype in our European cohort suggests these variants are of relatively recent origin and therefore may be population-specific [1,89,90].

While the bootstrapping and clustering in an independent cohort demonstrated that the subtypes were reproducible, the Jaccard scores were relatively modest, with only the reproductive subtype yielding a mean Jaccard coefficient $\bar{\gamma}_C > 0.6$. At least part of this outcome was likely due to the fact that all traits were fitted to a normal distribution using an inverse normal transformation prior to clustering. This transformation was done in order to prevent outliers from dictating cluster formations but also likely resulted in greater cluster overlap. Consequently, the metabolic and reproductive clusters we identified appear to represent opposite ends of a phenotypic spectrum with imperfect delineation. This spectrum, however, aligns with the known pathophysiology of PCOS and is bolstered by our genetic association findings. Our approach, therefore, appears to be a more reliable way of identifying subgroups of PCOS cases who have been noted in the literature [91] but have previously been defined using only a single trait like BMI [92–95] or by diagnostic criteria that do not reflect the genetic heterogeneity of the disorder [22]. Perhaps future studies that use clustering to identify reproductive and metabolic subtypes in PCOS can omit nondistinguishing traits such as DHEAS and T in an effort to reduce noise and improve subtype delineation and reproducibility.

Our study provides support for the hypothesis that PCOS is in fact a heterogeneous disorder with different underlying biological mechanisms. As a consequence, grouping women with PCOS under a single diagnosis may be counterproductive because distinct disease subtypes will likely benefit from different interventions.

In conclusion, using an unsupervised clustering approach featuring quantitative hormonal and anthropometric data, we identified reproductive and metabolic subtypes of PCOS that appeared to have distinct genetic architecture. The genomic loci that were significantly associated with either of these subtypes include a number of new, to our knowledge, highly plausible PCOS candidate genes. Moreover, our results demonstrate that precise phenotypic delineation, resulting in more homogeneous subsets of affected individuals, can be more powerful and informative than increases in sample size for genetic association studies. Our findings

indicate that further study into the genetic heterogeneity of PCOS is warranted and could lead to a transformation in the way PCOS is classified, studied, and treated.

## Supporting information

**S1 Checklist. STREGA checklist.**
(DOCX)

**S1 Table. GWAS cohorts used in cluster analysis.** The cohorts from the Hayes, Urbanek, and colleagues PCOS GWAS and corresponding numbers of samples that were included in the clustering analysis are shown by GWAS cohort, adapted from Hayes, Urbanek, and colleagues. [19]: Table 1 and Supplemental Data Tables 9 and 10. GWAS, genome-wide association study; PCOS, polycystic ovary syndrome.
(DOCX)

**S2 Table. Age and BMI distributions for subjects excluded from cluster analysis.** Median age and BMI values are shown with the 25th and 75th percentiles for the subjects included in the cluster analysis and for those from the same GWAS cohorts who were excluded because of missing quantitative trait data. Distributions were compared using unpaired Wilcoxon rank–sum tests. P-values are unadjusted. BMI, body mass index; GWAS, genome-wide association study.
(XLSX)

**S3 Table. Assays used to measure quantitative traits.** Assays used to measure quantitative trait levels are listed by trait, then by site and methodology combination. Unless otherwise noted, kits were used per the manufacturer's instructions. *Calibrated to WHO 1st International Standard #95/560. [a]Diagnostic Products Corporation (DPC) (Los Angeles, CA, USA) [Note: In April 2006, DPC was acquired by Siemens Medical Solutions USA, Inc. (Malvern, PA, USA)]. [b]Diagnostic Systems Laboratories, Inc. (DSL) (Webster, TX, USA) [Note: In October 2005, DSL was acquired by Beckman Coulter (Brea, CA, USA)]. [c]Siemens Medical Solutions USA, Inc. (Malvern, PA, USA). [d]Beckman Coulter, Inc. (Brea, CA, USA) [Note: In June 2011, Beckman Coulter was acquired by Danaher Corporation (Washington, DC, USA)]. [e]Analox Instruments Ltd. (London, UK). [f]Linco Research, Inc. (St. Charles, MO, USA). [g]American Laboratory Products Company (ALPCO) (Salem, NH, USA). BWH, Brigham and Women's Hospital; DHEAS, dehydroepiandrosterone sulfate; ELISA, enzyme-linked immunosorbent assay; FSH, follicle-stimulating hormone; GO, Glucose; G0, fasting glucose; HMC, Pennsylvania State Milton S. Hershey Medical Center; IRMA, immunoradiometric assay; I0, fasting insulin; LH, luteinizing hormone; NU, Northwestern University; RIA, radioimmunoassay; SHBG, sex hormone binding globulin; T, testosterone; UVA, University of Virginia.
(XLSX)

**S4 Table. Quantitative traits in genotyped PCOS cohort by cluster.** Median trait values are shown with 25th and 75th percentiles for each clustering subtype. Details for each assay method are provided in S3 Table. BMI, body mass index; DHEAS, dehydroepiandrosterone sulfate; FSH, follicle-stimulating hormone; G0, fasting glucose; I0, fasting insulin; LH, luteinizing hormone; N, total number; PCOS, polycystic ovary syndrome; SHBG, sex hormone-binding globulin; T, testosterone.
(XLSX)

**S5 Table. Subtypes of Stage 1 GWAS samples.** Subtypes are provided for each of the 555 Stage 1 GWAS samples included in the clustering analysis according to their dbGaP SUBJIDs. dbGaP, database of Genotypes and Phenotypes; GWAS, genome-wide association study;

SUBJID, subject ID.
(TXT)

## Acknowledgments

We thank the NIH Cooperative Multicenter Reproductive Medicine Network (https://www.nichd.nih.gov/research/supported/rmn) for recruiting some of the women with PCOS who participated who participated in the GWAS of Hayes and Urbanek and colleagues [19] and whose genotype data were used in this study. We also thank the following investigators for recruiting some of the control women who participated in the GWAS of Hayes and Urbanek and colleagues [19] and whose genotype data were used in this study: Dimitrios Panidis, MD, PHD (Aristotle University of Thessaloniki, Greece); Mark O. Goodarzi (Cedars-Sinai Medical Center, Los Angeles, CA, USA); Corrine K. Welt, MD (University of Utah School of Medicine, Salt Lake City, UT, USA; formerly of Massachusetts General Hospital, Boston, MA, USA); Ahmed H. Kissebah (deceased, Medical College of Wisconsin, Milwaukee, WI, USA); Ricardo Azziz, MD (State University of New York, NY, USA; formerly of University of Alabama at Birmingham, AL, USA); and Evanthia Diamanti-Kandarakis, MD, PhD (University of Athens Medical School, Greece).

## Author Contributions

**Conceptualization:** Matthew Dapas, M. Geoffrey Hayes, Andrea Dunaif.

**Data curation:** Matthew Dapas, Frederick T. J. Lin, Ryan Sisk.

**Formal analysis:** Matthew Dapas, Frederick T. J. Lin, Ryan Sisk.

**Funding acquisition:** Andrea Dunaif.

**Investigation:** Matthew Dapas, Richard S. Legro.

**Methodology:** Matthew Dapas, Girish N. Nadkarni.

**Project administration:** Ryan Sisk, Andrea Dunaif.

**Resources:** Richard S. Legro.

**Software:** Matthew Dapas.

**Supervision:** Margrit Urbanek, M. Geoffrey Hayes, Andrea Dunaif.

**Validation:** Matthew Dapas.

**Visualization:** Matthew Dapas.

**Writing – original draft:** Matthew Dapas.

**Writing – review & editing:** Matthew Dapas, Girish N. Nadkarni, Margrit Urbanek, M. Geoffrey Hayes, Andrea Dunaif.

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
