## [Decision Letter · Decision Letter 0]

28 Jan 2020

Dear Dr. Dapas,

Thank you very much for submitting your manuscript "Phenotypic clustering reveals distinct subtypes of polycystic ovary syndrome with novel genetic associations" (PMEDICINE-D-19-03066) for consideration at PLOS Medicine. 

Your paper was evaluated by a senior editor and discussed among all the editors here. It was also discussed with an academic editor with relevant expertise, and sent to four independent reviewers, including a statistical reviewer. The reviews are appended at the bottom of this email and any accompanying reviewer attachments can be seen via the link below:

[LINK]

In light of these reviews, I am afraid that we will not be able to accept the manuscript for publication in the journal in its current form, but we would like to consider a revised version that addresses the reviewers' and editors' comments. Obviously we cannot make any decision about publication until we have seen the revised manuscript and your response, and we plan to seek re-review by one or more of the reviewers. 

Of particular importance, in order for us to consider a revised submission we request that you please upload the relevant code to GitHub or a similar accessible repository, including an accompanying README document, as noted by Reviewer 2.

We expect to receive your revised manuscript by Feb 11 2020 11:59PM. Please email us (plosmedicine@plos.org) if you have any questions or concerns.

We look forward to receiving your revised manuscript. 

Sincerely,

Caitlin Moyer, Ph.D.

Associate Editor 

PLOS Medicine

plosmedicine.org

1. Title: Please revise your title according to PLOS Medicine's style. Your title must be nondeclarative and not a question. It should begin with main concept if possible. "Effect of" should be used only if causality can be inferred, i.e., for an RCT. Please place the study design ("A randomized controlled trial," "A retrospective study," "A modelling study," etc.) in the subtitle (ie, after a colon). 

We suggest: “Novel genetic associations of distinct subtypes of polycystic ovary syndrome: An unsupervised hierarchical clustering analysis” or similar.

2. Data Availability: PLOS Medicine requires that the de-identified data underlying the specific results in a published article be made available, without restrictions on access, in a public repository or as Supporting Information at the time of article publication, provided it is legal and ethical to do so. If the data are not freely available, please include an appropriate contact (web or email address) for inquiries (please note that this cannot be a study author). Please see the policy at: 

http://journals.plos.org/plosmedicine/s/data-availability

and FAQs at 

http://journals.plos.org/plosmedicine/s/data-availability#loc-faqs-for-data-policy

3. Prospective data analysis plan: Did your study have a prospective protocol or analysis plan? Please state this (either way) early in the Methods section. If your study has an associated prospective protocol or analysis plan, please include the file as supporting information, and reference it in the Methods.

c) In either case, changes in the analysis—including those made in response to peer review comments—should be identified as such in the Methods section of the paper, with rationale.

4. Abstract: Methods and Findings: Please include some information on the population, years during which the study took place, and the study’s main outcome measures.

5. Abstract: Conclusions: Line 58: The term “stable” in this sentence is unclear, please revise.

6. Author Summary: At this stage, we ask that you include a short, non-technical Author Summary of your research to make findings accessible to a wide audience that includes both scientists and non-scientists. The Author Summary should immediately follow the Abstract in your revised manuscript. This text is subject to editorial change and should be distinct from the scientific abstract. Please see our author guidelines for more information: https://journals.plos.org/plosmedicine/s/revising-your-manuscript#loc-author-summary

7. Methods: Thank you for including your Ethics Statement on the manuscript submission form (“The study was approved by the Institutional Review Board of Northwestern University Feinberg School of Medicine (#STU00008096). All subjects gave written informed consent before study.”), can you please also include this information in the methods? 

8. Methods: Please present a table of summary/demographic information for the various cohorts- the information that is in S1 Table should be moved to the main text.

9. Results: “PCOS Subtypes”: For the percentages reported here, please present numerators and denominators.

10. Discussion: Please present and organize the Discussion as follows: a short, clear summary of the article's findings; what the study adds to existing research and where and why the results may differ from previous research; strengths and limitations of the study; implications and next steps for research, clinical practice, and/or public policy; one-paragraph conclusion.

11. Figure 2: The small inset frequency histogram is difficult to read, particularly the Y axis. Can you please provide any description of the clustering relationships among the traits displayed on the left side of the chart?

12. Figure 5: Panel “a” is difficult to read, in particular the text of the histogram is very small (and appears to be missing an X axis label). Please increase the size of the text. Please define all abbreviations used in the figure legend.

13. Figure 6: Please increase the size of the Y axis labels. Please increase the size of the QQ plots, they are too small to read.

14. Figure 7: The left-hand side axis is cutting off one of the CI bars for the rs55762028 risk allele.

15. Figure 8, 9, 10: Please increase the text size on the graphs.

16. Table 1: Please define abbreviations for “Chr” and “Mb” in the legend.

17. References: Please use square brackets for in-text citations, like this: [1].

18. Checklist: Please report your study according to the relevant guideline (STREGA or STROBE may be most appropriate), which can be found here: http://www.equator-network.org/

Please include the completed STROBE/STREGA checklist as Supporting Information. When completing the checklist, please use section/paragraph numbers rather than page numbers. Please add the following statement, or similar, to the Methods: "This study is reported as per the Strengthening the Reporting of Observational Studies in Epidemiology (STROBE) guideline (S1 Checklist)."

Comments from the reviewers:

Reviewer #1: In" Phenotypic clustering reveals distinct subtypes of polycystic ovary syndrome with novel genetic associations" the authors describe a two step genetic and hormonal analysis to bettern understand underlying genetic changes in women with two different phenotypes of PCOS. The authors initially performed a Unsupervised hierarchical cluster analysis of anthropometric, reproductive, and metabolic traits in a genotyped discovery cohort of 893 PCOS cases and identified a reproductive (High LH and SHBG) and a metabolic phenotype. These clusters were confirmed in a secondary cohort and a bootstrapped sample. GWAS was then performed in many of these samples and in healthy controls and found 4 alleles identified with the reproductive and 1 with the metabolic phenotype. Additional study was performed from a cohort of familial clustered PCOS individuals.

Overall the study is vell-designed, well-powered and well-presented. The results add to literature of trying to understand a genetic basis for PCOS.

Abstract: The results for the unclustered 293 aren't presented? The 73 family cluster results were a surprise- should have been listed with other 2 cohorts at tope of methods.

Figure 1, if possible to put reproductive, metabolic and indeterminate at the top of the heatmap, it would make it easier for the reader to interpret.

Since total T reflect the influence of SHBG, which is different between the groups, it would be of interest to see if FAI is a better indicator, ie Fig 3

Axses on Figure 8 need to be larger

Reviewer #2: The manuscript investigates the presence of molecular subtypes in polycystic ovary syndrome (PCOS). The authors apply unsupervised learning to phenotypic quantitative traits and identify three distinct clusters, which they call "reproductive", "metabolic" and "intermediate". The first two are subsequently fed to a Genome-Wide Association Study (GWAS) to look for distinctive genetic variants underlying the corresponding population groups. Results suggest genetic heterogeneity and distinct disease subtypes that get classified under the same PCOS umbrella.

Overall, this is a very well-written paper with sound methodology and fair presentation. The statistical analyses are well motivated, and conclusions are evaluated for robustness using standard tools like bootstrap resampling, multiple hypothesis testing correction, replication in a separate validation cohort, and reporting of confidence regions. The few places where the tests are underpowered (i.e., GWAS), the results are presented through a realistic lens. There are only a few relatively minor criticisms of the work.

First, there is no accompanying code, which makes the analyses irreproducible. My suggestion is to create a code repository (e.g., on GitHub) and upload the scripts used for clustering, GWAS and machine learning analyses. A README detailing how to access the datasets used by the code must be included.

There is a slight disconnect between Introduction and the presented findings. Examples provided during the opening (lines 65-71) give an impression that the authors are investigating genetically heterogeneous subtypes that give rise to a convergent phenotype. However, the first step of the analysis is a phenotypic separation (i.e., divergence) of samples into clusters. This doesn't invalidate the study, but I think a minor change in the opening angle could be appropriate.

In several places, the analysis considers projections of data onto the first k principal components. This includes outlier detection (line 190) and batch adjustment (line 200). The authors need to show that the number of components considered (two and three, respectively) captures the sufficient amount of signal in the data. The standard way of doing this is via Horn's Parallel Analysis (doi:10.1007/bf02289447) where the data is randomly permuted along each dimension, and the resulting principal component variance of the scrambled data is compared to the original.

I would like to see more justification for why the hierarchical clustering was collapsed to only three subtypes. To me, it seems that there's a fourth cluster within the "Intermediate" subtype, characterized by high levels of ovarian theca cell testosterone (T) production. This is visually supported by the distinct pattern in the heatmap (Fig 2), as well as a strong alignment of T with the second principal component in both cohorts (Figs 4 and 5). If Jaccard coefficient was used to make this decision, then the authors need to also report coefficients associated with the two-cluster solution ("reproductive" and "metabolic + intermediate") and the four-cluster solution ("reproductive", "metabolic", "T-high intermediate" and "T-low intermediate"), highlighting that the three-cluster solution yields an inflection point.

Please include a short statement about whether there was general agreement between the various classification methods (lines 220-225). Large discrepancy in performance may indicate that subtype classifications are not robust, and a further look at the data properties may be required to motivate the proper method choice (e.g., there may be strong nonlinear relationships between input variables that warrant the use of random forests).

Lastly, I agree with the authors that the GWAS analyses may be underpowered. Given that significant signal from previous GWAS was not observed in any of the subtypes (line 392), and that there is very limited replication of hits in the Stage 2 cohort (Table 1), I feel that a number of the hits identified may actually be spurious (in particular, BMPR1B/UNC5C and CDH10). However, the authors readily admit that the GWAS findings are preliminary and provide some Hi-C data and literature-backed justification for why and how the identified variants may regulate PCOS pathways.

Minor comments:

-Table S1 provides only the summary statistics. In line with the earlier comment about reproducibility, it would be helpful to include detailed instructions on how the raw data was accessed.

-Please provide reasoning behind the r^2 threshold of 0.8 on line 194. It seems a bit high given the sample size.

-Double-check the significance of FSH in Fig3. Visually, it seems the median of each group falls within the interquartile range of the other group. I am surprised this leads to a significant distinction, especially after Bonferroni correction.

-Figure 3 caption: missing definition of T.

Reviewer #3: The ms by Dapas et al suggests that the diagnostic rubric "polycystic ovarian syndrome" is heterogeneous in clinical presentation and that the phenotypic heterogeneity has genetic underpinnings. The notion that PCOS is not one diagnostic category is an overdue concept. The authors of the present ms have presented data to support the notion that anovulation due to PCOS is more complex that previously appreciated. 

Accepting that the analytic methodology is sufficiently rigorous to support the authors' analysis and conclusions, parsing PCOS suggests that clinicians need to refine treatment approaches. Women with PCOS are poorly served by being lumped into a single diagnostic entity as doing so then suggests that all women with PCOS need similar interventions. 

Reviewer #4: The authors of this paper have undertaken a study to see whether they could identify different phenotypical PCOS subgroups with a different genetic background. This hypothesis was driven by the fact that a recent GWAS didn't show any differences between women with PCOS diagnosed according to different diagnostic criteria. Indeed this suggests that diagnostic criteria currently used are not adequately identifying phenotypes that might matter in a clinical sense. In order to do so they first undertook a unsupervised cluster analysis in order to identify different subtypes of PCOS. Thereafter they did a GWAS on the different subtypes and identified new loci associated with the newly defined phenotypes. They also did a similar analysis in in a family based PCOS cohort and showed that a rare DENND1A variant was significantly more often found in the reproductive phenotype compared to the metabolic phenotype.

The study is well designed and the authors have to be congratulated with this large and innovative approach. This is a complex and laborious endeavour. Methods seem to be appropriate and statistics that were applied, although complex, are similarly adequate. The manuscript reads well and the arguing is consistent and sound. The paper is well written and the figures and tables are easy to read and appropriately summarise the findings of the study.

I have only one major issue that is:

That is, although the Jaccard indices, depicting that the phenotypes are really distinct, are not that high to claim that with the certainty of how it is written down in the results. In fact there is only one value that is really above 0.6 i.e. the one from the phenotype that is associated with the reproductive trait. This should be mentioned in more detail in the discussion. Although this constitutes a limitation the study is still reporting valid an interesting data and provides a novel view of how to define PCOS.

There is also a few minor issues such as:

NIH diagnosed women with PCOS constitute a subset of the whole population diagnosed according to the Rotterdam criteria. This should be mentioned in the M&M section.

Was there any effort made to compare (in terms of general characteristics such as age, BMI etc.) the group of 893 individuals and the ones from whom not a complete set quantitative trait date were available. May be it is also worthwhile to show how many out of each cohort were selected for this analysis.

I was wondering how many women were using OCP or other hormones until 3 months prior to actual phenotyping. Knowing that some effects of especially OCP's might last longer than 3 months this might have impacted on the data. For instance Testosterone levels and SHBG levels might very well be suppressed for quite some time after discontinuing OCP's. If only a limited number of women would have been using OCP's until recently this would make the data more solid than they already are. This should be mentioned too in case these data are available.

[LINK]

---

## [Decision Letter · Decision Letter 1]

17 Apr 2020

Dear Dr. Dapas,

Thank you very much for re-submitting your manuscript "Distinct subtypes of polycystic ovary syndrome with novel genetic associations: an unsupervised, phenotypic clustering analysis" (PMEDICINE-D-19-03066R1) for review by PLOS Medicine.

I have discussed the paper with my colleagues and the academic editor and it was also seen again by two reviewers. I am pleased to say that provided the remaining editorial and production issues are dealt with we are planning to accept the paper for publication in the journal.

[LINK]

We look forward to receiving the revised manuscript by Apr 24 2020 11:59PM. 

Sincerely,

Caitlin Moyer, Ph.D.

Associate Editor 

PLOS Medicine

plosmedicine.org

Requests from Editors:

1.The Data Availability Statement (DAS) requires revision. For each data source used in your study: 

a) If the data are freely or publicly available, note this and state the location of the data: within the paper, in Supporting Information files, or in a public repository (include the DOI or accession number). Specifically, please provide the direct link for the Phase I genotype data in the database of Genotypes and Phenotypes (dbGaP).

For the remaining (sequencing/array) aggregate data: 

Please include an appropriate contact (web or email address) for inquiries (again, this cannot be a study author). When you say “Investigators may contact the Site PIs from Hayes & Urbanek et al. [18] if they are interested in collaborating on a project that requires use of quantitative trait data.” please note that the present study authors may not serve as the contact points for data access.

2. Abstract: Methods and Findings: Please include appropriate summary demographic information regarding the individuals included in the analyses.

3. Abstract: Conclusions: Please revise or remove the final sentence of the abstract to be more congruent with the main findings of the study: “This study demonstrates how precise phenotypic delineation can be more powerful than increases in sample size for genetic association studies.” Please mention only specific implications of the study substantiated by the results.

4. Author Summary: “Why was this study done?”: We suggest revising the final bullet point to “Elucidating the genetic mechanisms of PCOS could result in improved diagnosis and treatment.” to remove implications of causality.

5 .Author Summary: “What do these findings mean?”: Please revise the first bullet point to remove causal language- we suggest: “Our results suggest that there are distinct forms of PCOS that are associated with different underlying biological mechanisms.”

6. Methods: Line 198: Please include the analyses of age and BMI compared between cases included and excluded due to missing quantitative trait data, as PLOS does not permit "data not shown.” (This could be presented in a supporting information file).

7. Methods: Please provide a summary table to go along with the results of Figure 3 (summary data on quantitative phenotypic trait data).

8. Figure 1: We request that you please remove this figure from the paper, and re-number the rest of the figures and in-text figure references, accordingly.

9. Figure 5: In the legend where you refer to “unclassified (grey) clusters” please clarify whether this is the same as the “indeterminate” clusters.

10. Figure 6: The inset graphs are too small to read easily, please enlarge the fonts for readability. In the legend, please include the panel for “indeterminate” PCOS: “Manhattan plots for (a) reproductive, (b) metabolic, and (c) indeterminate PCOS subtypes.”

11. Figure 7: Please define the abbreviation “OR” in the figure legend.

12. Figure 8: Please increase the size of the smaller fonts, as they are difficult to read.

13. Figure 12: Please define abbreviations for PC1/PC2/PCA; DHEAS, LH, FSH, G0, I0, BMI,SHBG in the figure legend. Please explain that the markers indicate affected women and the bold marker outlines represent carriers.

14. Supporting Information Tables: Please provide titles and legends for each individual table and figure in the Supporting Information.

Comments from Reviewers:

Reviewer #2: The authors carefully and thoroughly addressed all my concerns. I have no further requests and gladly recommend this manuscript for publication.

Reviewer #4: My concerns have been met appropriately. So I don't have any major or minor concerns regarding this paper.

[LINK]

---

## [Editor Report · Decision Letter 2]

13 May 2020

Dear Dr. Dapas, 

On behalf of my colleagues and the academic editor, Dr. Jenny Myers, I am delighted to inform you that your manuscript entitled "Distinct subtypes of polycystic ovary syndrome with novel genetic associations: an unsupervised, phenotypic clustering analysis" (PMEDICINE-D-19-03066R2) has been accepted for publication in PLOS Medicine. 

PRODUCTION PROCESS

PRESS

PROFILE INFORMATION

Thank you again for submitting the manuscript to PLOS Medicine. We look forward to publishing it. 

Best wishes, 

Caitlin Moyer, Ph.D.

Associate Editor 

PLOS Medicine

plosmedicine.org